# TOWARDS FAIR CLASSIFICATION AGAINST POISONING ATTACKS

## ABSTRACT

Fair classification aims to stress the classification models to achieve the equality (treatment or prediction quality) among different sensitive groups. However, fair classification can be under the risk of poisoning attacks that deliberately insert malicious training samples to manipulate the trained classifiers' performance. In this work, we study the poisoning scenario where the attacker can insert a small fraction of samples into training data, with arbitrary sensitive attributes as well as other predictive features. We demonstrate that the fairly trained classifiers can be greatly vulnerable to such poisoning attacks, with much worse accuracy & fairness trade-off, even when we apply some of the most effective defenses (originally proposed to defend traditional classification tasks). As countermeasures to defend fair classification tasks, we propose a general and theoretically guaranteed framework which accommodates traditional defense methods to fair classification against poisoning attacks. Through extensive experiments, the results validate that the proposed defense framework obtains better robustness in terms of accuracy and fairness than representative baseline methods.

## 1 INTRODUCTION

Data poisoning attacks (Biggio et al., 2012; Chen et al., 2017; Steinhardt et al., 2017) have brought huge safety concerns for machine learning systems that are trained on data collected from public resources (Konečnỳ et al., 2016; Weller & Romney, 1988). For example, the studies (Biggio et al., 2012; Mei & Zhu, 2015; Burkard & Lagesse, 2017; Steinhardt et al., 2017) have shown that an attacker can inject only a small fraction of fake data into the training pool of a classification model and intensely degrade its accuracy. As countermeasures against data poisoning attacks, there are defense methods (Steinhardt et al., 2017; Diakonikolas et al., 2019) which can successfully identify the poisoning samples and sanitize the training dataset.

Recently, in addition to model safety, people have also paid significant attention to fairness. They stress that machine learning models should provide the "equalized" treatment or "equalized" prediction quality among groups of population (Hardt et al., 2016; Agarwal et al., 2018; Donini et al., 2018; Zafar et al., 2017). Since the fair classification problems are human-society related, it is highly possible that training data is provided by humans, which can cause high accessibility for adversarial attackers to inject malicious data. Therefore, fair classification algorithms are also prone to be threatened by poisoning attacks. Since fair classification problems have distinct optimization objectives & optimization processes from traditional classification, a natural question is: *Can we protect fair classification from data poisoning attacks? In other words, are existing defenses sufficient to defend fair classification models?*

To answer these questions, we first conduct a preliminary study on Adult Census Dataset to explore whether existing defenses can protect fair classification algorithms (see Section 3.2). In this work, we focus on representative defense methods including k-NN Defense (Koh et al., 2021) and SEVER (Diakonikolas et al., 2019). To fully exploit their vulnerability to poisoning attacks, we introduce a new attacking algorithm *F-Attack*, where the attacker aims to cause the failure of fair classification. In detail, by injecting the poisoning samples, the attacker aims to mislead the trained classifier such that it cannot achieve good accuracy, or not satisfy the fairness constraints. From the preliminary results, we find that both k-NN defense and SEVER will have an obvious accuracy or fairness degradation after attacking. Moreover, we also compare F-Attack with one of the strongest poisoning attacks Min-Max Attack Steinhardt et al. (2017) (which is devised for traditional classification). The result demonstrates that our proposed F-Attack has a better attacking effect compared to Min-Max Attack. In

conclusion, our preliminary study highlights the vulnerability of fair classification against poisoning attacks, especially against F-Attack.

In this paper, we further propose a defense framework, Robust Fair Classification (RFC), to improve the robustness of fair classification against poisoning attacks. Different from existing defenses, our method aims to scout abnormal samples from each individual sensitive subgroup in each class. To achieve this goal, RFC first applies the similar strategy as the works (Diakonikolas et al., 2017; 2019), to find abnormal data samples which significantly deviate from the distribution of other (clean) samples. Based on a theoretical analysis, we verify that RFC can exclude more poisoning samples than clean samples in each step. Moreover, to further avoid removing too many clean samples, we introduce an *Online-Data-Sanitization* process: in each iteration, we remove a possible poisoning set from a single subgroup of a single class and test the retrained models' performance on a clean validation set. This helps us locate the subgroup which contains the most poisoning samples. Through extensive experiments on two benchmark datasets, Adult Census Dataset and COMPAS, we validate the effectiveness of our defense. Our key contributions are summarized as:

- We devise a strong attack method to poison fair classification and demonstrate the vulnerability of fair classification under the protection of traditional defenses to poisoning attacks.
- We propose an efficient, and principled framework, Robust Fair Classification (RFC). Extensive experiments and theoretical analysis demonstrate the effectiveness and reliability of the proposed framework.

## 2 PROBLEM STATEMENT AND NOTATIONS

In this section, we formally define the setting of our studied problem and necessary notations.

**Fair Classification.** In this paper, we focus on the classification problems which incorporate group-level fairness criteria. First, let $x \subseteq \mathbb{R}^d$ be a random vector denoting the (non-sensitive) features, with a label $y \in \mathcal{Y} = \{Y_1, ..., Y_m\}$ with $m$ classes, a sensitive attribute $z \in \mathcal{Z} = \{Z_1, Z_2, ..., Z_k\}$ with $k$ groups. Let $f(X, w)$ represent a classifier with parameters $w \in \mathcal{W}$. Then, a fair classification problem can be defined as:

$$\min_w \mathbb{E}\Big[l(f(x, w), y)\Big] \text{ s.t. } g_j(w) \leq \tau, \forall j \in \mathcal{Z} \tag{1}$$

where the function $\mathbb{E}[l(\cdot)]$ is the expected loss on test distribution, and $\tau$ is the unfairness tolerance. The constraint function $g_j(w) = \mathbb{E}\Big[h(w, x, y)|z = j\Big]$ represents the desired fairness metric for each group $j \in \mathcal{Z}$. For example, in binary classification problems, we use $f(x, w) > 0$ to indicate a positive classification outcome. Then, $h(w, x, y) = \mathbf{1}(f(x, w) > 0) - \mathbb{E}\Big[\mathbf{1}(f(x, w) > 0)\Big]$ refers to equalized positive rates in the *equalized treatment* criterion (Mehrabi et al., 2021). Similarly, $h(w, x, y) = \mathbf{1}(f(x, w) > 0|y = \pm1) - \mathbb{E}\Big[\mathbf{1}(f(x, w) > 0|y = \pm1)\Big]$ is for equalizing true / false positive rates in the *equalized odds* (Hardt et al., 2016). Given any training dataset $D$, we define the empirical loss function as $L(D, w)$ as the average loss value of the model, and $g_j(D, w)$ is the empirical fairness constraint function.

In our paper, we assume that the clean training samples are sampled from the true distribution $\mathcal{D}$ following the density $\mathbb{P}(x, y, z)$. We also use $\mathcal{D}^{u,v}$ to denote the distribution of (clean) samples given by $y = Y_u$ and $z = Z_u$, which has a density $\mathbb{P}(x, y, z|y = Y_u, z = Z_v)$.

**Poisoning Attack ($\epsilon$-poisoning model).** In our paper, we consider the poisoning attack following the scenario. Given a fair classification task, the poisoned dataset is generated as follows: first, $n$ clean samples $D_C = \{(x_i, y_i, z_i)\}_{i=1}^n$ are drawn from $\mathcal{D}$ to form the clean training set. Then, an *adversary* is allowed to insert an $\epsilon$ fraction of $D_C$ with arbitrary choices of $D_P = \{(x_i, y_i, z_i)\}_{i=1}^{\epsilon n}$. We can define such a poisoned training set $D_C \cup D_P$ as $\epsilon$-*poisoning model*.

## 3 FAIR CLASSIFICATION IS VULNERABLE TO POISONING ATTACKS

In this section, we first introduce the algorithm of our proposed attack F-Attack under the $\epsilon$-poisoning model. Then, we conduct empirical studies to evaluate the robustness of fair classification algorithms (and popular defenses) against F-Attack and baseline attacks.

### 3.1 F-ATTACK: POISONING ATTACKS FOR FAIR CLASSIFICATION

Given a specific fair classification task, we consider that the attacker aims to contaminate the training set, such that applying existing algorithms cannot successfully fulfill the fair classification goal. Note that for fair classification tasks, both accuracy and fairness are the desired properties and they always have strong tension in practice Menon & Williamson (2018). Therefore, in our attack, we consider misleading the training algorithms such that at least one of the two criteria is unsatisfied. Formally, we define the attacker's objective as Eq.(2), where the attacker inserts a poisoning set $D_P$ with size $\epsilon n$ in the feasible injection space $\mathcal{F}$ to achieve:

$$\max_{D_P \subseteq \mathcal{F}} \quad \mathbb{E}\Big[l(f(x, w^*), y)\Big] \quad \text{s.t.} \quad w^* = \arg\min_{w \in \mathcal{H}_{fair}} L(D_C \cup D_P, w). \tag{2}$$

It means for the classifier $w^*$ that trained on $D_C \cup D_p$ and has a low empirical loss $L(D_C \cup D_p, w^*)$, if it falls in the space $\mathcal{H}_{fair}$, it will have a large expected loss (on the test distribution $\mathcal{D}$). Here, $\mathcal{H}_{fair}$ is the space of models (with a limited norm) that satisfy the fairness criteria on clean distribution $\mathcal{D}$. To have a closer look at Eq.(2), we discuss case by case. Suppose we obtain $w^*$ by fair classification on the set $D_C \cup D_P$, there are cases:

1. $w^* \notin \mathcal{H}_{fair}$: The fairness criteria (on the test set) is not satisfied.
2. $w^* \in \mathcal{H}_{fair}$: Since $w^*$ is trained on $D_C \cup D_P$ to have low $L(D_c \cup D_p, w^*)$, $w^*$ will have high test error.

For each case, the model $w^*$ will either have an unsatisfactory accuracy or unsatisfactory fairness. Next, we simplify the objective and constraints in Eq.(2) to transform it into a solvable problem. We first conduct relaxations of the objective in Eq.(2) (similar to the works (Steinhardt et al., 2017; Koh et al., 2021)):

$$\mathbb{E}\Big[l(f(X, w), Y)\Big] \overset{(i)}{\approx} L(D_C, w) \overset{(ii)}{\leq} L(D_C, w) + \epsilon L(D_P, w) = (1 + \epsilon)L(D_C \cup D_P, w)$$

Specifically, approximation (i) holds if clean training data $D_C$ has sufficient samples and is close to the test distribution $\mathcal{D}$, and model $w$ is appropriately regularized. The upper bound (ii) holds because of the non-negativity of loss values. The upper bound (ii) can be tight if the fraction of poisoning samples $\epsilon$ is small. Thus, we transfer Eq.(2) to a bi-level optimization problem between $w$ and $D_P$. If the model $f(\cdot)$ and loss $l(\cdot)$ are convex, we can further swap them to get a min-max form as:

$$\max_{D_P \subseteq \mathcal{F}} \min_{w \in \mathcal{H}_{fair}} \quad L(D_C \cup D_P, w) \rightarrow \min_{w \in \mathcal{H}_{fair}} \max_{D_P \subseteq \mathcal{F}} \quad L(D_C \cup D_P, w) \tag{3}$$

Our proposed F-Attack is to solve Eq.(3) which is shown in Algorithm 1. It solves a saddle point problem to alternatively find the worst attack points $(x, y, z)$ w.r.t the current model and then update the model in the direction of the attack point. In detail, in Step (1), given the current model $w$, we solve the inner maximization problem to maximize $L(D_C \cup D_P, w)$. It is equal to finding sample $(x, y, z)$ with the maximal loss:

$$\max_{D_P \subseteq \mathcal{F}} L(D_C \cup D_P, w) = L(D_C, w) + \epsilon \cdot \max_{(x,y,z) \in \mathcal{F}} l(f(x, w), y). \tag{4}$$

In Step (2), we update $w$ to minimize $L(D_C \cup D_P, w)$. Note that in Step (2), we should also constrain the model $w$ to fall into the fair model space $\mathcal{H}_{fair}$. Thus, when we update $w$ in Step (2), we also penalize the fairness violation of $w$. Here, we calculate the fairness violation as $(g_j(D_C, w) - \tau)^+$ (with weight parameter $\lambda > 0$) on the clean set $D_C$ to approximate the fairness violation on real data $\mathcal{D}$.

---

**Algorithm 1:** Algorithm of F-Attack

---

**Input** : Clean data $D_C$, number of poisoned samples $\epsilon n$, feasible set $\mathcal{F}$, Fairness constraint functions $g_j(\cdot)$ and tolerance $\tau_j$ ($j = 1, ..., |\mathcal{Z}|$), $\lambda > 0$, $\eta > 0$, warm-up steps $n_{burn}$.
**Output** : A poisoning set $D_P^*$
**for** $t = 1, ..., n_{burn} + \epsilon n$ **do**
    1. Solve inner maximization: $(x, y, z) = \arg\max_{(x,y,z) \in \mathcal{F}} l(f(x, w), y)$ and let $D_P = \epsilon n \times \{(x, y, z)\}$
    2. Solve outer minimization: $w = w - \eta \cdot \nabla_w((L(D_C \cup D_P, w) + \lambda \sum_j (g_j(D_C, w) - \tau)^+))$
    **if** $t > n_{burn}$ **then**
        | $D_P^* = D_P^* \cup \{(x, y, z)\}$
    **end**
**end**

---

## 3.2 PRELIMINARY STUDY ON ADULT CENSUS DATASET

**Adult Census Dataset.** In this subsection, we conduct an experiment on Adult Census Dataset Kohavi et al. (1996), to test whether F-Attack can poison fair classification methods and whether existing defense methods can resist F-Attack. Here, we focus on the fairness criteria: Equalized True Positive Rate (TPR) Hardt et al. (2016) between the genders, and we apply the constrained optimization method Donini et al. (2018) to train linear classifiers to fulfill the fair classification objective. It is worth mentioning that, this dataset contains many categorical features, such as marital-status, occupation, etc. For simplicity, we pre-process the dataset by transforming categorical features into a continuous space that is spanned by the first 15 principle directions of training (categorical) data. More details of the pre-processing procedure can be found in Appendix A.2.

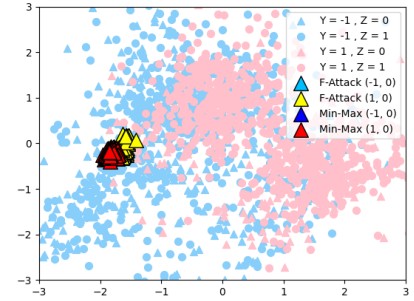

Figure 1: PCA visualization of Clean Samples and Poisoning Samples

**Defense Methods.** Besides naïve fair classification, we mainly consider two representative data-sanitization defenses, which are existing popular methods to defend against poisoning attacks:

- *k-NN Defense (Koh et al., 2021).* This method removes the samples that are far from their k nearest neighbors. In detail, the k-NN defense calculates the "abnormal" score as $q_i = ||x_i - x_i^{(k,y)}||_2$, where $x_i^{(k,y)}$ is the k-th nearest neighbor to sample $x_i$ in class $y$. In this paper, we set $k = 5$.

- *SEVER. Diakonikolas et al. (2019)* This method aims to find abnormal samples by tracing abnormal gradients. In each iteration, we first train a fair classifier $f$ (with fixed $\tau$) and calculate the gradient of loss w.r.t the weight $w$ for each training sample $(x_i, y_i)$, and get the normalized gradient matrix $Q = \left[ \nabla_w l(f(x_i, w), y_i) - \frac{1}{n} \sum_{j=1}^n \nabla_w l(f(x_j, w), y_j) \right]_{i=1,..,n}$. SEVER flags the samples with large "abnormal" score $q_i = (Q_i \cdot v)^2$ as abnormal samples, where $v$ is the top right singular vector of $Q$. Intuitively, the "abnormal" samples make a great contribution to the variation of the gradient matrix $Q$, which suggests their gradients can significantly deviate from the gradients of other samples.

**Results.** In our experiments, we insert 10% poisoning samples to the training set, and define the feasible injection set $\mathcal{F}$ to be $\{(x, y, z) : ||x - \mu_y|| \le d\}$, where $d$ is a fixed radius. This will constrain the inserted samples not too far from the center of their labeled class, to evade potential defense. Since $\mathcal{F}$ is not related to sensitive attribute $z$, during F-Attack, we generate poisoning samples with a fixed $z$ to be 0 (female) or 1 (male). During fairness training, we train multiple models with various hyperparameters to control the unfairness tolerance on the training set (following (Lamy et al., 2019)). Then, we report the test performance when it has the best validation performance (which considers both accuracy and fairness, see Section 4, Eq.(9) for more details). In Table 1, we report the performance[1] for the defense methods. From the result, we can see: all training methods have a significant performance degradation under F-Attack. For example, under F-Attack ($z = 0$), the SEVER defense has $\approx 4\%$ accuracy drop and $2\%$ fairness drop. This suggests that defenses such as SEVER and k-NN can be greatly vulnerable to poisoning attacks in fair classification. Moreover, we compare F-Attack with a baseline attack method Min-Max (Steinhardt et al., 2017; Koh et al., 2021), which is one of the strongest attacks for traditional classification. It also solves Eq.(3) but does not constrain $w \in \mathcal{H}_{fair}$. From Table 1, we can see that Min-Max has worse attacking performance than F-Attack, by causing slighter performance degradation. This result highlights the threat of F-Attack to fair classification.

Table 1: F-Attack vs. Min-Max on Adult Census Dataset

|  | **No Attack** | | **Min-Max (z = 0)** | | **Min-Max (z = 1)** | | **F-Attack (z = 0)** | | **F-Attack (z = 1)** | |
|---|---|---|---|---|---|---|---|---|---|---|
|  | Acc. | Fair. | Acc. | Fair. | Acc. | Fair. | Acc. | Fair. | Acc. | Fair. |
| **No Defense.** | 0.811 | 0.962 | 0.801 | 0.958 | 0.799 | 0.851 | 0.768 | 0.956 | 0.793 | 0.803 |
| **k-NN.** | 0.794 | 0.952 | 0.681 | 0.936 | 0.680 | 0.904 | 0.655 | 0.951 | 0.695 | 0.895 |
| **SEVER.** | 0.812 | 0.969 | 0.798 | 0.958 | 0.797 | 0.967 | 0.773 | 0.942 | 0.772 | 0.943 |

---

[1]We report the "goodness of fairness" as $1 -$ Unfair, i.e., $1 - |\text{TPR}(z = 0) - \text{TPR}(z = 1)|$ in Table 1.

**Discussion.** To have a deeper understanding on the behavior of F-Attack, in Figure 1, we visualize the clean samples and poisoning samples (via F-Attack and Min-Max Attack) in a 2-dim projected space (via PCA). From the figure, we can see that: compared to Min-Max Attack (red), the samples obtained by F-Attack (yellow) have a smaller distance to the clean samples in their labeled class $y = 1$, although they are constrained in the same feasible injection set $\mathcal{F}$. It is because F-Attack aims to find samples with maximal loss (Eq.(4)) for fair classifiers, so the generated samples do not have the maximal loss for traditional classifiers. Thus, the poisoning samples from F-Attacks are closer to their labeled class. This fact helps explain why F-Attack is more insidious than Min-Max Attack under the detection of traditional defenses, such as SEVER. In Appendix A.4, we further conduct experiments on Adult Census Dataset focused on a Neural Network based classification model, which demonstrates that the idea of F-Attack has potential to can be adapted to DNN models.

## 4 ROBUST FAIR CLASSIFICATION (RFC)

Motivated by studies in Section 3, new defenses are desired to protect fair classification against poisoning attacks, especially against F-Attack. In this section, we first introduce a novel defense framework called Robust Fair Classification (RFC), and we provide a theoretical study to further understand the mechanism of RFC. In Section 5, we conduct empirical studies to validate the robustness of RFC in practice.

### 4.1 ROBUST FAIR CLASSIFICATION (RFC)

Based on the discussion in Section 3, F-Attack can evade traditional defenses such as SEVER and k-NN Defense, because the generated poisoning samples are close to the clean data of their labeled class. However, we assume that they may deviate from the distribution of clean samples in their labeled subgroup (the data distribution $\mathcal{D}^{y,z}$ given $y$ and $z$). Refer to the Figure 2, which shows the location of poisoning samples generated via F-Attack ($z = 0$) and clean samples in subgroup ($y = 1, z = 0$) in the 2D projected space. It suggests that the poisoning samples greatly contaminate the information of distribution $\mathbb{P}(x, y, z | y = 1, z = 0)$ in the training data. Thus, the injected poisoniend samples will not only confuse the original prediction task from $x$ to $y$, but also greatly disturb the fairness constraints (Eq.(1)).

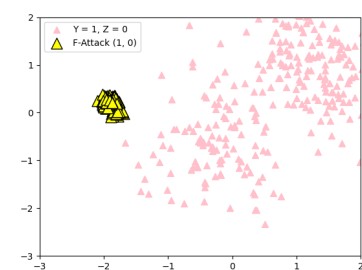

Figure 2: PCA visualization in $(1, 0)$

This observation motivates us to propose a new defense method that can scout abnormal samples from each individual subgroup in each class. Next, we will introduce the details of our proposed defense RFC.

**Centered Data Matrix & Poisoning Score.** Our method shares a similar high-level idea as (Diakonikolas et al., 2017; 2019), to find data points that systematically deviate from the distribution of other (clean) samples. Specifically, in our method, given a (poisoned) dataset $D$, we repeatedly scout the poisoning samples from each subgroup $D(x, y, z | y = Y_u, z = Z_v)$, where we use $(u, v)$ to denote the index of each subgroup and class. In the later parts, we use $D^{u,v}$ to denote the samples in $D(x, y, z | y = Y_u, z = Z_v)$ for simplicity. Then, we define:

$$Q^{u,v} = \left[ x_i - \frac{1}{n_{u,v}} \sum_{j=1}^{n_{u,v}} x_j \right]_{(x_i, y_i, z_i) \in D^{u,v}}, \tag{5}$$

to be the *centered data matrix* of training samples $D^{u,v}$ and $n_{u,v}$ is the size of the set $D^{u,v}$.

For each $Q^{u,v}$, the top right singular vector $\mathcal{V}^{u,v}$ of $Q^{u,v}$ is the direction which explains the variation of the data distribution in $(Y_u, Z_u)$. Similar to the studies in Diakonikolas et al. (2017; 2019), we conjecture: the poisoning samples are deviated from clean samples, so they will take the major responsibility for the variation of the data matrix $Q^{u,v}$. In this way, they will have high alignments with the direction of $\mathcal{V}^{u,v}$. Thus, we define the *Poisoning Score* $q^{u,v}$ for each training sample $(x_i, y_i, z_i)$ in the $D^{u,v}$ as:

$$q^{u,v}(x_i) = Q_i^{u,v} \cdot (\mathcal{V}^{u,v})^T. \tag{6}$$

Notably, the poisoned samples are likely to have the same or opposite direction with the top right singular vector $\mathcal{V}^{u,v}$, but the poisoning samples should share the same direction. Thus, in our method, we define two *Proposed*

*Poisoning Sets* for each $(u, v)$, so that one of the two sets is likely to have poisoning samples:

$$\mathcal{P}_+^{u,v} = \{x_i \,|\, q^{u,v}(x_i) > \gamma_+, q^{u,v}(x_i) > 0\}; \ \ \mathcal{P}_-^{u,v} = \{x_i \,|\, -q^{u,v}(x_i) > \gamma_-, q^{u,v}(x_i) < 0\}; \quad (7)$$

In Eq.(7), we set the $\gamma_+$ (or $\gamma_-$) to be the $q$-th ($q = 90$) percentile of the given all Poisoning Scores (or negative poisoning Scores) in $D^{u,v}$, so that each proposed poisoning set only contains a small portion of $D^{u,v}$. In practice, we will repeatedly test whether removing the proposed poisoning sets can help improve the retrained model's performance (both accuracy and fairness) on a clean validation set. It helps to decide whether the proposed poisoning set contains poisoning samples. In Section 4.2, we will further conduct a theoretical analysis to show that the poisoning samples are more likely to have higher poisoning scores.

**Fair Classification by Excluding Poisoning Set.** Finally, given a (poisoned) training set $D$ as well as the proposed poisoning sets, we can keep retraining fair classifiers by excluding proposed poisoning sets:

$$\begin{aligned}
w_+ &= \arg\min_{w \in \mathcal{W}} L(f(X, w), Y; D \setminus \mathcal{P}_+^{u,v}) \ \text{s.t.} \ g_j(D \setminus \mathcal{P}_+^{u,v}; w) \leq \tau_j, \forall j \in \mathcal{Z} \\
w_- &= \arg\min_{w \in \mathcal{W}} L(f(X, w), Y; D \setminus \mathcal{P}_-^{u,v}) \ \text{s.t.} \ g_j(D \setminus \mathcal{P}_-^{u,v}; w) \leq \tau_j, \forall j \in \mathcal{Z}
\end{aligned} \quad (8)$$

In practice, our proposed RFC method repeatedly proposes potential poisoning sets for each individual subgroup in each class $D^{u,v}$ until finding the best poisoning set among all choices. The Algorithm 2 provides the detailed introduction of the procedure of RFC, which is an *Online Data Sanitization* process. Specifically, during each iteration of RFC, for each $D^{u,v}$, we first calculate the poisoning scores and the proposed poisoning sets (Step (1)&(2)). Then, we remove the proposed poisoning set from the dataset $D$, and conduct fair classification on $D$ (Step (3)). In Step (4), we evaluate the retrained classifier on a clean separated validation set and find the best-proposed poisoning set which results in the highest validation performance. Notably, we measure the validation performance by considering both the accuracy and fairness criteria, by defining:

$$ValScore = \Pr.(f(x, w) = y) - \sum_{j \in \mathcal{Z}} \lambda \cdot (g_j(D_{Val}, w) - \tau)^+, \quad (9)$$

where $\tau$ is a unfairness tolerance threshold and $\lambda$ is a positive number (we set $\lambda = 3$ in this paper). The second term penalizes the models if some subgroups' unfairness violation is over $\tau$. Finally, we remove the proposed poisoning set which results in the highest *ValScore* and conduct the next round of searching (Step (6)).

---

**Algorithm 2:** Robust Fair Classification (RFC) - An Online Data Sanitization Algorithm

---

**Input** : An $\epsilon$-poisoning model with dataset $D = \{(x_i, y_i, z_i)\}_{i=1,2,...,n}$, Iterations $T$ of RFC.
**Output :** A fair classifier
**while** $t \leq T$ **do**
    **for** $Y_u \in \mathcal{Y}$, $Z_v \in \mathcal{Z}$ **do**
        1. Get the *Centered Data Matrix* $Q^{u,v}$ and *Poisoning Score* $q_i^{u,v}$ following Eq.(5) and Eq.(6)
        2. Get the *Proposed Poisoning Sets*: $\mathcal{P}_+^{u,v}$ and $\mathcal{P}_-^{u,v}$ following Eq.(7)
        3. Conduct fair classification by removing *Proposed Poisoning Set*, via Eq.(8) and get $w_\pm^{u,v}$.
        4. Record the performance to get *ValScore* on a separated (clean) validation set for each $w_\pm^{u,v}$.
    **end**
    5. Get the best proposed poisoning set $\mathcal{P}^*$ which achieves the highest *ValScore* across all $u, v$ and $\mathcal{P}_\pm^{u,v}$.
    6. Removing the best proposed poisoning set from $D$ and set $D = D \setminus \mathcal{P}^*$
**end**

---

Remarkably, it is also worth mentioning that the framework RFC is also possible to be extended to various model architectures, such as Deep Neural Networks (DNNs), for robust fair classification. For example, we can apply the *Energy-based Out-of Distribution Detection* Liu et al. (2020) to find the abnormal samples from each $D^{u,v}$. Then, we follow a similar manner as RFC to propose poisoning sets and conduct fair classification. We will leave the study in DNNs for future exploration.

## 4.2 THEORETICAL ANALYSIS

In this subsection, we conduct a theoretical analysis to further help understand the behavior of RFC, especially to understand the role of calculating poisoning scores in finding poisoning samples. In particular, we consider a

simple theoretical setting where the clean samples from each group $\mathcal{D}^{u,v}$ follow a distinct Gaussian distribution $\mathcal{D}^{u,v} \sim \mathcal{N}(\mu^{u,v}, \Sigma^{u,v})$, with center and covariance matrix $(\mu^{u,v}, \Sigma^{u,v})$. In the following theorem, we will show: when there are (poisoning) samples that deviate from the clean samples of $\mathcal{D}^{u,v}$, by having a center $\mu$ which is far from the center of clean samples, they will have larger squared *poisoning scores* (Eq.6) than clean samples. Thus, the proposed poisoning sets (Eq.(7)) are likely to contain more poisoning samples than clean samples. For simplicity, we use $\mathcal{D}$ and $\mathcal{N}(\mu, \Sigma)$ to denote the clean distribution of a given group $D^{u,v}$.

**Theorem 4.1.** *Suppose that a set of "clean" samples $\mathcal{S}_{good}$ with size $n$ are i.i.d sampled from distribution $\mathcal{N}(\mu, \Sigma)$, where $\Sigma \preceq \sigma^2 I$. There is a set of "bad" samples $\mathcal{S}_{bad}$ with size $n_p = n/K, K > 1$ and center $||\mu_p - \mu||_2 = d, d = \gamma \cdot \sigma$. Then, the average squared poisoning scores of clean samples and bad samples have a relationship:*

$$\mathbb{E}_{i \in \mathcal{S}_{bad}} \left[ q^2(x_i) \right] - \mathbb{E}_{i \in \mathcal{S}_{good}} \left[ q^2(x_i) \right] \geq \left( \frac{K-1}{K+1} \cdot \gamma^2 - (K+1) \right) \sigma^2$$

Theorem 4.1 suggests that the difference between the average (squared) poisoning scores of $\mathcal{S}_{bad}$ and $\mathcal{S}_{good}$ is controlled by $\gamma$ and the sample ratio $K$. Since $K > 1$, if $\gamma^2 > \frac{(K+1)^2}{K-1}$ (which suggests the poisoning samples are sufficiently far from clean distribution), we can get the conclusion that the difference is positive. Thus, removing samples with the highest positive (or lowest negative) poisoning scores (as Eq.(7)) will help to eliminate more poisoning samples than clean samples. If $\gamma^2$ is small, the poisoning samples are close to the true distribution, which will cause the poisoning samples to have limited influence on the model performance. The detailed proof of Theorem 4.1 is deferred to Appendix A.1. In our algorithm of RFC, we alternatively check each proposed poisoning set and see whether removing it helps improve the retrained models' performance. This will also avoid removing too many clean samples. In Appendix A.5, we provide extra discussion on the precision of the detection of RFC, which demonstrates that RFC will not remove too many clean samples in practice.

## 5 EXPERIMENT

### 5.1 EXPERIMENTAL SETUP

In this section, we conduct comprehensive experiments to validate the effectiveness of our proposed attack and defense, in two benchmark datasets, Adult Census Dataset and COMPAS Dataset. In this part, we only consider Equalized True Positive Rate (TPR) between different sensitive subgroups, which is optimized via the fair classification method (Donini et al., 2018). When applying (Donini et al., 2018), we train multiple models with various hyperparameters to control the unfairness tolerance on the training set (following (Lamy et al., 2019)). Then, we report the test performance when it has the best validation performance (which considers both accuracy and fairness, see Section 4, Eq.(9)). In Appendix A.3, we provide additional results for a different type of fairness "Equalized Treatment", and a different fair classification method (Zafar et al., 2017). The implementation can be found at `https://anonymous.4open.science/r/f_attack-4017/`.

**Attacks**: We consider that the training set can be contaminated by: Label Flipping (Paudice et al., 2018), and Sensitive Attribute Flipping (Wang et al., 2020). We also consider the attack methods, Min-Max and F-Attack, which are introduced in Section 3. Notably, for each method, we assume the poisoning samples are constrained in the sample feasible injection set $\mathcal{F} = \{(x, y, z) : ||x - \mu_y|| \leq d\}$, which limit the poisoning samples' distance to the class center. Thus, for Min-Max and F-Attack, we assign the generated samples to have a pre-defined sensitive attribute $z = 0$ or $z = 1$. Furthermore, we introduce an additional attack method "F-Attack*" which has the same algorithm with F-Attack but have a different feasible injection set: $\mathcal{F} = \{(x, y, z) : ||x - \mu_{y,z}|| \leq d\}$, where $\mu_{y,z}$ is the center of the group $\mathcal{D}^{y,z}$. Because this feasible injection set is related to the sensitive attribute $z$, we don't need to pre-define $z$ during F-attack*. Remarkably, this attack aims to test the robustness of RFC, because the major goal of RFC is to find samples in each group $D^{y,z}$ which are far from $\mu_{y,z}$. Thus, F-Attack* is possible to evade RFC by constraining the poisoning samples' distance to $\mu_{y,z}$. In Appendix A.3, we also report the performance of all attacks & defenses under different choices of radius $d$.

**Baseline Defenses.** To validate the effectiveness of RFC, we include baseline defense methods: (1) the naive method which does not apply any defense strategies; (2) SEVER (Diakonikolas et al., 2019), which are representative defenses for traditional classification tasks. We apply (Donini et al., 2018) on the sanitized dataset by SEVER. In addition, we also include (3) Roh et al. (2020), which is a method to defend fair classification methods against label flipping attacks. It leverages adversarial training strategy Zhang et al. (2018); and (4) the method (Wang et al., 2020) applies Distributional Robust Opitmization (DRO) to improve robustness when labels

Table 2: RFC & Baseline Methods' Performance against Poisoning Attacks on Adult Census Dataset.

| | No Attack | | Label Flip(10%) | | Label Flip(20%) | | Attr. Flip(10%) | | Attr. Flip(20%) | |
|---|---|---|---|---|---|---|---|---|---|---|
| | Acc. | Fair. | Acc. | Fair. | Acc. | Fair. | Acc. | Fair. | Acc. | Fair. |
| No Defense. | **0.817** | **0.963** | 0.805 | 0.966 | 0.800 | **0.968** | 0.812 | 0.945 | 0.804 | 0.951 |
| SEVER. | 0.812 | 0.960 | 0.805 | 0.967 | **0.803** | 0.960 | **0.813** | 0.937 | 0.803 | 0.944 |
| Wang. | 0.809 | 0.958 | 0.793 | 0.970 | 0.779 | 0.961 | 0.809 | 0.945 | **0.813** | 0.935 |
| Roh. | 0.805 | 0.948 | 0.798 | 0.937 | 0.788 | 0.939 | 0.800 | 0.950 | 0.795 | 0.943 |
| RFC. | 0.811 | 0.959 | **0.807** | **0.973** | 0.796 | 0.966 | 0.805 | **0.967** | 0.802 | **0.965** |

| ($\epsilon = 10\%$) | Min-Max($z = 0$) | | Min-Max($z = 1$) | | F-Attack($z = 0$) | | F-Attack($z = 1$) | | F-Attack* | |
|---|---|---|---|---|---|---|---|---|---|---|
| | Acc. | Fair. | Acc. | Fair. | Acc. | Fair. | Acc. | Fair. | Acc. | Fair. |
| No Defense. | 0.801 | 0.969 | 0.797 | 0.864 | 0.769 | 0.966 | 0.796 | 0.804 | 0.799 | 0.799 |
| SEVER. | 0.800 | 0.960 | 0.795 | 0.954 | 0.778 | 0.945 | 0.773 | 0.937 | 0.772 | **0.956** |
| Wang. | 0.782 | **0.976** | 0.783 | **0.968** | 0.779 | 0.956 | 0.791 | 0.948 | 0.792 | 0.945 |
| Roh. | 0.780 | 0.963 | 0.782 | 0.961 | 0.765 | **0.959** | 0.766 | **0.956** | 0.776 | 0.944 |
| RFC. | **0.802** | 0.967 | **0.811** | 0.946 | **0.803** | 0.950 | **0.808** | 0.952 | **0.809** | 0.951 |

| ($\epsilon = 15\%$) | Min-Max($z = 0$) | | Min-Max($z = 1$) | | F-Attack($z = 0$) | | F-Attack($z = 1$) | | F-Attack* | |
|---|---|---|---|---|---|---|---|---|---|---|
| | Acc. | Fair. | Acc. | Fair. | Acc. | Fair. | Acc. | Fair. | Acc. | Fair. |
| No Defense. | 0.792 | 0.961 | 0.787 | 0.837 | 0.686 | 0.953 | 0.775 | 0.779 | 0.788 | 0.760 |
| SEVER. | 0.792 | 0.959 | 0.783 | **0.978** | 0.765 | 0.952 | 0.755 | 0.909 | 0.765 | 0.927 |
| Wang. | 0.777 | **0.970** | 0.779 | 0.965 | 0.738 | 0.955 | 0.762 | **0.960** | 0.711 | **0.955** |
| Roh. | 0.726 | 0.948 | 0.748 | 0.957 | 0.722 | 0.962 | 0.764 | 0.929 | 0.774 | 0.929 |
| RFC. | **0.801** | 0.954 | **0.796** | 0.941 | **0.800** | **0.963** | **0.795** | 0.947 | **0.802** | 0.945 |

and sensitive attributes are contaminated. For baseline methods, we report their performance with the choice of hyperparameter that achieves the optimal *ValScore* (Eq.( 9)) on a clean validation set.

## 5.2 EXPERIMENTAL RESULTS

**Adult Census Dataset.** We first show the results in Adult Census Dataset in Table 2. To further guarantee that the comparison between different defenses is fair, we use a balanced clean training dataset where each class has an equal number of samples since the baseline methods such as SEVER can be affected by class imbalance. Under this dataset, we set the desired fairness criteria to be $|\text{TPR}(z = 0) - \text{TPR}(z = 1)| < 0.05$. In Table 2, we also mark the cases (with brown color) when the algorithms output models with much poorer fairness than the desired fairness. From Table 2, we can see that RFC can achieve good accuracy & fairness among different types of dataset contamination. Especially, under strong attacks such as F-Attack, the accuracy and fairness are only slightly degraded after injecting poisoning samples. However, the baseline methods such as Wang et al. (2020) and Roh et al. (2020), will have a clear performance (especially accuracy) degradation under F-Attack. Notably, the attack method F-Attack*, has similar attacking performance as F-Attack ($z = 1$). It is because under this dataset, F-Attack* also generates samples that have $z = 1$.

**COMPAS Dataset.** In COMPAS dataset (Brennan et al., 2009), we consider the same type of fairness criteria, which is Equalized TPR. In this dataset, we consider that the equity is desired among races, which are "Caucasian ($z = 0$), African-American ($z = 1$) and Hispanic ($z = 2$)", and follow the similar preprocessing procedure as that in Adult Census Dataset. In this dataset, the number of samples in the group "Hispanic" is much smaller than the other two groups. Thus, we only consider to inject poisoning samples to $z = 0$ or $z = 1$. In Table 3, we report the performance of our studied attacks and defense, and we use $1 - \max_{j \in \mathcal{Z}} |\text{TPR}(z = j) - \hat{\text{TPR}}|$ to measure the "goodness" of fairness, where $\hat{\text{TPR}}$ is the averaged TPR in the whole dataset. During training, we set the desired fairness criteria to be $\max_{j \in \mathcal{Z}} |\text{TPR}(z = j) - \hat{\text{TPR}}| \leq 0.15$. From the result in Table 3, we can see that RFC is the only method that can consistently preserve the model accuracy and fairness after there are poisoning samples injected into the dataset.

## 6 RELATED WORKS

**Poisoning Attacks.** In this section, we introduce related work and discuss how this work differs from prior studies. Data poisoning attacks (Biggio et al., 2012) refer to the scenario that models are threatened by adversaries who insert malicious training samples, in order to take control of the trained model behavior (Li et al., 2020;

Table 3: RFC & Baseline Methods' Performance against Poisoning Attacks on COMPAS Dataset.

| | No Attack | | Label Flip(10%) | | Label Flip(20%) | | Attr. Flip(10%) | | Attr. Flip(20%) | |
|---|---|---|---|---|---|---|---|---|---|---|
| | Acc. | Fair. | Acc. | Fair. | Acc. | Fair. | Acc. | Fair. | Acc. | Fair |
| **No Defense.** | 0.656 | **0.865** | 0.655 | 0.866 | 0.664 | 0.836 | 0.677 | 0.839 | 0.665 | 0.847 |
| **SEVER.** | 0.650 | 0.854 | **0.677** | 0.833 | 0.674 | 0.835 | 0.674 | 0.835 | **0.667** | 0.835 |
| **Wang.** | **0.662** | 0.847 | 0.650 | 0.869 | 0.631 | **0.863** | 0.643 | 0.861 | 0.659 | **0.862** |
| **Roh.** | 0.654 | 0.851 | 0.646 | **0.891** | 0.663 | 0.823 | 0.621 | 0.834 | 0.615 | 0.845 |
| **RFC.** | 0.661 | 0.850 | 0.676 | 0.859 | **0.682** | 0.841 | **0.685** | 0.850 | **0.667** | 0.856 |

| ($\epsilon = 10\%$) | Min-Max(z = 0) | | Min-Max(z = 1) | | F-Attack(z = 0) | | F-Attack(z = 1) | | F-Attack* | |
|---|---|---|---|---|---|---|---|---|---|---|
| | Acc. | Fair. | Acc. | Fair. | Acc. | Fair. | Acc. | Fair. | Acc. | Fair |
| **No Defense.** | 0.649 | 0.845 | 0.645 | 0.837 | 0.665 | 0.832 | 0.630 | 0.860 | 0.661 | 0.826 |
| **SEVER.** | 0.634 | 0.865 | 0.630 | 0.859 | 0.634 | 0.845 | 0.627 | 0.840 | 0.664 | 0.826 |
| **Wang.** | 0.648 | 0.851 | 0.620 | 0.855 | 0.639 | **0.865** | 0.624 | **0.871** | 0.644 | **0.868** |
| **Roh.** | 0.644 | **0.880** | 0.631 | 0.865 | 0.659 | **0.865** | 0.633 | 0.825 | 0.640 | 0.858 |
| **RFC.** | **0.656** | 0.863 | **0.661** | 0.834 | **0.668** | 0.853 | **0.667** | 0.866 | **0.673** | 0.845 |

| ($\epsilon = 15\%$) | Min-Max(z = 0) | | Min-Max(z = 1) | | F-Attack(z = 0) | | F-Attack(z = 1) | | F-Attack* | |
|---|---|---|---|---|---|---|---|---|---|---|
| | Acc. | Fair. | Acc. | Fair. | Acc. | Fair. | Acc. | Fair. | Acc. | Fair |
| **No Defense.** | **0.676** | 0.802 | 0.644 | 0.863 | **0.665** | 0.802 | 0.630 | 0.860 | 0.642 | 0.831 |
| **SEVER.** | 0.621 | **0.880** | 0.620 | **0.880** | 0.585 | **0.890** | 0.606 | **0.887** | 0.663 | 0.845 |
| **Wang.** | 0.645 | 0.865 | 0.631 | 0.852 | 0.606 | 0.842 | 0.611 | 0.829 | 0.624 | 0.841 |
| **Roh.** | 0.655 | 0.847 | 0.628 | 0.877 | 0.625 | 0.865 | 0.631 | 0.849 | 0.621 | **0.877** |
| **RFC.** | **0.676** | 0.852 | **0.659** | 0.843 | 0.653 | 0.848 | **0.645** | 0.847 | **0.672** | 0.841 |

Shafahi et al., 2018). In this work, we concentrate on the untargeted poisoning attacks (Biggio et al., 2012; Koh et al., 2021) where the attacker aims to degrade the overall performance of the trained model. To defend against poisoning attacks, well-established methods (Wilcox, 2011; Rubinstein et al., 2009; Steinhardt et al., 2017; Diakonikolas et al., 2019; Tao et al., 2021; Wang et al., 2021b) are proposed to efficiently and effectively defend against poisoning attacks in various scenarios. This paper is within the scope of linear classification problems and we leave the studies in DNN models for future work.

**Fair Classification.** Fairness issues have recently drawn much attention from the community of machine learning. Fairness issues for common classification problems can be generally divided into two categories: (1) Equalized treatment Zafar et al. (2017) (or "Statistical Rate"); and (2) Equalized prediction quality (Hardt et al., 2016). For classification models to satisfy these fairness criteria, popular methods including (Zafar et al., 2017; Donini et al., 2018; Agarwal et al., 2018) solve constrained optimization problems, and (Zhang et al., 2018) apply adversarial training (Madry et al., 2017) method.

**Comparison to Prior Works.** There are recent works that try to test the robustness of fair classification methods by manipulating their training set. They also proposed possible strategies to defend the perturbations. For example, the works (Wang et al., 2020; Lamy et al., 2019; Celis et al., 2021a;b) consider injecting naturally / adversarially generated noise only on sensitive attributes. Another line of researches Roh et al. (2020); Wang et al. (2021a) considers the vulnerability of fairness training to (coordinated) label-flipping attacks (Paudice et al., 2018). As countermeasures to defend against their proposed perturbations, representative works such as (Roh et al., 2020) proposed an adversarial training framework (Zhang et al., 2018), to train the model to distinguish clean samples and poisoning samples, while preserving the model fairness. The work (Wang et al., 2020) solves robust optimization problems by assigning soft sensitive attributes. In our work, in terms of attack, we consider a stronger attacker because he/she can insert sophisticatedly calculated features and sensitive attributes, to fully exploit the vulnerability of fairness training methods.

## 7 CONCLUSION

In this work, we study the problem of poisoning attacks on fair classification problems. We propose a strong attack method that can evade the defense of most existing methods. Then, we propose an effective strategy to greatly improve the robustness of fair classification methods. In the future, we aim to examine if our findings can be generalized to other machine learning tasks, and other machine learning models, such as Deep Neural Networks (DNNs).

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

# A APPENDIX

## A.1 PROOF OF THEOREM

In this part, we provide the detailed proof of Theorem 4.1 in Section 4.

**Theorem A.1** (Recall Theorem 4.1)**.** *Suppose a set of "clean" samples $\mathcal{S}_{good}$ with size $n$ are i.i.d sampled from distribution $\mathcal{N}(\mu, \Sigma)$, where $\Sigma \preceq \sigma^2 I$. There is a set of "bad" samples $\mathcal{S}_{bad}$ with size $n_p = n/K, K > 1$ and center $||\mu_p - \mu||_2 = d, d = \gamma \cdot \sigma$. Then, the average squared poisoning scores of clean samples and bad samples have the relationship:*

$$\mathbb{E}_{i \in \mathcal{S}_{bad}} \left[ q^2(x_i) \right] - \mathbb{E}_{i \in \mathcal{S}_{good}} \left[ q^2(x_i) \right] \geq \left( \frac{K-1}{K+1} \cdot \gamma^2 - (K+1) \right) \sigma^2$$

*Proof.* We denote $\mathcal{S}_{good}$ is the set of clean samples, $\mathcal{S}_{bad}$ is the set of bad samples, and the union of clean samples and bad samples form the whole set $\mathcal{S}$. In the later part, to distinguish between the centers of each set, we use $\mu_g$, $\mu_p$ and $\mu_s$ to denote the center of clean samples, bad samples and the whole set.

First, it is easy to know that $\mu_g$, $\mu_p$ and $\mu_s$ are in a same line. Thus, given $n_g : n_p = K : 1$, we have the relationship of center distance: $d = ||\mu_g - \mu_p|| = (1 + K)||\mu_g - \mu_s|| = ((K + 1)/K)||\mu_p - \mu_s||$. In the following, we will study the squared poisoning score in the whole group $\mathcal{S}$. Given any unit vector $\mathcal{V}$ is the top right singular vector of the centered data matrix of $\mathcal{S}$:

$$\mathbb{E}_{i \in \mathcal{S}} \left[ (\mathcal{V} \cdot (x - \mu_s))^2 \right] = \mathbb{E}_{i \in \mathcal{S}} \left[ (\mathcal{V} \cdot (x - \mu_p) + \mathcal{V} \cdot (\mu_p - \mu_s))^2 \right]$$

$$= \mathbb{E}_{i \in \mathcal{S}} \left[ (\mathcal{V} \cdot (x - \mu_p))^2 \right] + \mathbb{E}_{i \in \mathcal{S}} \left[ 2(\mathcal{V} \cdot (x - \mu_p)(\mu_p - \mu_s)^T \cdot \mathcal{V}^T \right] + \mathbb{E}_{i \in \mathcal{S}} \left[ (\mathcal{V} \cdot (\mu_p - \mu_s))^2 \right]$$

$$= \mathbb{E}_{i \in \mathcal{S}} \left[ (\mathcal{V} \cdot (x - \mu_p))^2 \right] - (\mathcal{V} \cdot (\mu_p - \mu_s))^2$$

Note that $\mathcal{V}$ is the top right singular vector of the centered data matrix $X_s - \mu_s$, we choose $v' = (\mu_p - \mu_s)/||(\mu_p - \mu_s)||$, which is the unit vector that has the same direction with $(\mu_p - \mu_s)$. We get:

$$\mathbb{E}_{i \in \mathcal{S}} \left[ (\mathcal{V} \cdot (x - \mu_s))^2 \right] \geq \mathbb{E}_{i \in \mathcal{S}} \left[ (v' \cdot (x - \mu_p))^2 \right] - ||\mu_p - \mu_s||^2$$

For the first term in the right hand side of the inequality above:

$$(K + 1) \cdot \mathbb{E}_{i \in \mathcal{S}} \left[ (v' \cdot (x - \mu_p))^2 \right] \geq K \cdot \mathbb{E}_{i \in \mathcal{S}_{good}} \left[ (v' \cdot (x - \mu_p))^2 \right]$$

because the poisoning scores are all positive. Then, we have:

$$(K + 1) \cdot \mathbb{E}_{i \in \mathcal{S}} \left[ (v' \cdot (x - \mu_p))^2 \right] \geq K \cdot \mathbb{E}_{i \in \mathcal{S}_{good}} \left[ (v' \cdot (x - \mu_p))^2 \right]$$

$$= K \cdot \mathbb{E}_{i \in \mathcal{S}_{good}} \left[ (v' \cdot (x - \mu_g) + v' \cdot (\mu_g - \mu_p))^2 \right]$$

$$= K \cdot \left( v' \cdot (X_g - \mu_g)(X_g - \mu_g)^T \cdot v'^T + v' \cdot (\mu_g - \mu_p)(\mu_g - \mu_p)^T \cdot v'^T \right)$$

$$\geq K \cdot (0 + ||\mu_g - \mu_p||^2)$$

The first term is larger than 0 because of the semi-definite property of the matrix $(X_g - \mu_g)(X_g - \mu_g)^T$, the second term is because $v'$ has the same direction with $(\mu_p - \mu_s)$ (because $\mu_p$, $\mu_g$ and $\mu_s$ are in the same line). Therefore, we get the average poisoning score of the whole set $\mathcal{S}$:

$$\mathbb{E}_{i \in \mathcal{S}} \left[ (\mathcal{V} \cdot (x - \mu_s))^2 \right] \geq \frac{K}{K+1} \cdot ||\mu_g - \mu_p||^2 - ||\mu_p - \mu_s||^2 = \frac{K}{(1+K)^2} \cdot d^2$$

In the following, we will calculate the average squared poisoning score in the good set $\mathcal{S}_{good}$.

$$\mathbb{E}_{i \in \mathcal{S}_{good}} \left[ (\mathcal{V} \cdot (x - \mu_p))^2 \right] = \mathcal{V} \cdot (X_g - \mu_g)(X_g - \mu_g)^T \cdot \mathcal{V}^T + \mathcal{V} \cdot (\mu_g - \mu_s)(X_g - \mu_s)^T \cdot \mathcal{V}^T$$

$$\leq \sigma^2 + ||\mu_g - \mu_s||^2 \leq \sigma^2 + (\frac{1}{K+1} \cdot d)^2$$

Based on previous calculation about the average score of whole set and good set, we can get the average squared poisoning score in the bad set $\mathcal{S}_{bad}$:

$$\mathbb{E}_{i \in \mathcal{S}_{bad}} \left[ (\mathcal{V} \cdot (x - \mu_p))^2 \right] = (1 + K)\mathbb{E}_{i \in \mathcal{S}} \left[ (\mathcal{V} \cdot (x - \mu_p))^2 \right] - K\mathbb{E}_{i \in \mathcal{S}_{good}} \left[ (\mathcal{V} \cdot (x - \mu_p))^2 \right]$$

$$\geq \frac{K^2}{(K+1)^2} \cdot d^2 - K\sigma^2$$

Table 4: RFC & Baseline Methods' Performance on Adult Census Dataset, for Equalized Treatment

| | No Attack | | Label Flip(10%) | | Label Flip(20%) | | Attr. Flip(10%) | | Attr. Flip(20%) | |
| | Acc. | Fair. | Acc. | Fair. | Acc. | Fair. | Acc. | Fair. | Acc. | Fair |
|---|---|---|---|---|---|---|---|---|---|---|
| No Defense. | 0.795 | 0.808 | 0.787 | 0.823 | 0.784 | 0.834 | 0.797 | 0.800 | 0.802 | 0.800 |
| SEVER. | 0.772 | 0.829 | 0.754 | 0.823 | 0.757 | 0.801 | 0.785 | 0.795 | 0.781 | 0.790 |
| RFC. | 0.781 | 0.803 | 0.791 | 0.820 | 0.799 | 0.830 | 0.788 | 0.806 | 0.783 | 0.823 |

| $(\epsilon = 10\%)$ | Min-Max(z = 0) | | Min-Max(z = 1) | | F-Attack(z = 0) | | F-Attack(z = 1) | | F-Attack* | |
| | Acc. | Fair. | Acc. | Fair. | Acc. | Fair. | Acc. | Fair. | Acc. | Fair. |
|---|---|---|---|---|---|---|---|---|---|---|
| No Defense. | 0.779 | 0.834 | 0.787 | 0.810 | 0.781 | 0.795 | 0.785 | 0.823 | 0.790 | 0.822 |
| SEVER. | 0.766 | 0.803 | 0.757 | 0.800 | 0.770 | 0.788 | 0.764 | 0.788 | 0.769 | 0.803 |
| RFC. | 0.778 | 0.834 | 0.788 | 0.804 | 0.781 | 0.812 | 0.794 | 0.799 | 0.793 | 0.803 |

| $(\epsilon = 15\%)$ | Min-Max(z = 0) | | Min-Max(z = 1) | | F-Attack(z = 0) | | F-Attack(z = 1) | | F-Attack* | |
| | Acc. | Fair. | Acc. | Fair. | Acc. | Fair. | Acc. | Fair. | Acc. | Fair. |
|---|---|---|---|---|---|---|---|---|---|---|
| No Defense. | 0.782 | 0.797 | 0.783 | 0.792 | 0.766 | 0.787 | 0.777 | 0.822 | 0.770 | 0.801 |
| SEVER. | 0.775 | 0.769 | 0.766 | 0.792 | 0.741 | 0.803 | 0.755 | 0.773 | 0.766 | 0.770 |
| RFC. | 0.786 | 0.806 | 0.794 | 0.812 | 0.776 | 0.815 | 0.790 | 0.822 | 0.786 | 0.823 |

and the difference between two averaged scores:

$$\mathbb{E}_{i \in \mathcal{S}_{bad}} - \mathbb{E}_{i \in \mathcal{S}_{good}} \geq \frac{k-1}{k+1} \cdot d^2 - (k+1)\sigma^2 = \left( \frac{K-1}{K+1} \cdot \gamma^2 - (K+1) \right) \sigma^2$$

$\square$

## A.2  MORE EXPERIMENTAL DETAILS

In this part, we provide additional experimental details such as the pre-process procedure.

**Adult Cenesus Dataset.** In this dataset, we have 5 numerical features "age, education-num, hours-per-week, capital-loss and capital gain"', and we also use categorical features such as "workclass, education, marital-stataus, occupation, relationship, race". For simplicity, we first transform the categoircal features into dummy variables and conduct Principle Component Analysis to project them to the space, which is spanned by the first 15 principle components. After projection, we normalize all 20 features by centering and standadizing. Under this dataseet, during the attacking of Min-Max and F-Attack, we assign the radius of the feasible injection set to be $d = 9.0$. In Appendix A.3, we provide the empirical results for more choices of $d$, i.e., $d = 6.0$.

**COMPAS Dataset.** In this dataset, we have the numerical features: "age, age_cat, juv_fel_count, juv_misd_count, juv_other_count, priors_count, days_b_screening_arrest, decile_score, c_jail_in, c_jail_out". We use "c_jail_out - c_jail_in" to get the number of days in jail and exclude c_jail_out, c_jail_in. We also have categorical features c_charge_degree and sex, and we use PCA to find the first two principle directions. Then, we standardize each feature. Under this dataset, during the attacking of Min-Max and F-Attack, we assign the radius of the feasible injection set to be $d = 9.0$. In Appendix A.3, we provide the empirical results for more choices of $d$, i.e., $d = 6.0$.

## A.3  MORE EXPERIMENTAL RESULTS

In this part, we provide additional empirical results to validate the effectiveness of our attack and defense method. In detail, we consider more settings about: (1) Different Type of Fairness Criteria, such as "Equalized Treatment"; (2) Different fair classification method, such as (Zafar et al., 2017), (3) Different choice of the radius of feasible injection space $d$, such as $d = 6.0$. Notably, in the all experiments in the main paper, we set $d$ with a fixed value $d = 9.0$. In this part, we provide another option when we choose a smaller $d = 6.0$. It is because larger $d$, i.e, $d > 9.0$ will make the poisoning samples easier to be detected by most defense methods. For fairness criteria under "Equalized Treatment" (Equalized Positive Rate (PR)) in Adult Census Dataset, we set the desired fairness criteria to be $|\text{PR}(z = 0) - \text{PR}(z = 1)| < 0.2$. For fairness criteria under "Equalized TPR" in Adult Census Dataset, we set the unfairness criteria to be $|\text{TPR}(z = 0) - \text{TPR}(z = 1)| < 0.05$. For fairness criteria under "Equalized TPR" and "Equalized Treatment" in COMPAS Dataset, we set the desired fairness criteria to be $\max_{j \in \mathcal{Z}} |\text{TPR}(z = j) - \hat{\text{TPR}}| \leq 0.15$ and $\max_{j \in \mathcal{Z}} |\text{PR}(z = j) - \hat{\text{PR}}| \leq 0.15$.

Table 5: RFC & Baseline Methods' Performance on Adult Census Dataset using (Zafar et al., 2017).

| | No Attack | | Label Flip(10%) | | Label Flip(20%) | | Attr. Flip(10%) | | Attr. Flip(20%) | |
|---|---|---|---|---|---|---|---|---|---|---|
| | Acc. | Fair. | Acc. | Fair. | Acc. | Fair. | Acc. | Fair. | Acc. | Fair. |
| No Defense. | 0.815 | 0.944 | 0.813 | 0.962 | 0.804 | 0.954 | 0.814 | 0.942 | 0.811 | 0.931 |
| SEVER. | 0.812 | 0.952 | 0.806 | 0.954 | 0.808 | 0.957 | 0.814 | 0.942 | 0.811 | 0.953 |
| RFC. | 0.814 | 0.964 | 0.808 | 0.956 | 0.801 | 0.951 | 0.802 | 0.963 | 0.802 | 0.944 |

| $(\epsilon = 10\%)$ | Min-Max$(z=0)$ | | Min-Max$(z=1)$ | | F-Attack$(z=0)$ | | F-Attack$(z=1)$ | | F-Attack* | |
|---|---|---|---|---|---|---|---|---|---|---|
| | Acc. | Fair. | Acc. | Fair. | Acc. | Fair. | Acc. | Fair. | Acc. | Fair. |
| No Defense. | 0.801 | 0.969 | 0.797 | 0.864 | 0.769 | 0.966 | 0.796 | 0.804 | 0.799 | 0.799 |
| SEVER. | 0.800 | 0.960 | 0.802 | 0.954 | 0.788 | 0.945 | 0.786 | 0.937 | 0.779 | 0.956 |
| RFC. | 0.802 | 0.967 | 0.811 | 0.946 | 0.803 | 0.950 | 0.808 | 0.952 | 0.809 | 0.951 |

| $(\epsilon = 15\%)$ | Min-Max$(z=0)$ | | Min-Max$(z=1)$ | | F-Attack$(z=0)$ | | F-Attack$(z=1)$ | | F-Attack* | |
|---|---|---|---|---|---|---|---|---|---|---|
| | Acc. | Fair. | Acc. | Fair. | Acc. | Fair. | Acc. | Fair. | Acc. | Fair. |
| No Defense. | 0.784 | 0.953 | 0.797 | 0.841 | 0.700 | 0.943 | 0.755 | 0.821 | 0.798 | 0.800 |
| SEVER. | 0.802 | 0.959 | 0.793 | 0.978 | 0.775 | 0.952 | 0.778 | 0.909 | 0.775 | 0.927 |
| RFC. | 0.799 | 0.944 | 0.799 | 0.952 | 0.796 | 0.950 | 0.801 | 0.951 | 0.803 | 0.946 |

Table 6: RFC & Baseline Methods' Performance on Adult Census Dataset when $d = 6$

| $(\epsilon = 10\%)$ | Min-Max$(z=0)$ | | Min-Max$(z=1)$ | | F-Attack$(z=0)$ | | F-Attack$(z=1)$ | | F-Attack* | |
|---|---|---|---|---|---|---|---|---|---|---|
| | Acc. | Fair. | Acc. | Fair. | Acc. | Fair. | Acc. | Fair. | Acc. | Fair. |
| No Defense. | 0.796 | 0.970 | 0.803 | 0.955 | 0.762 | 0.977 | 0.803 | 0.778 | 0.811 | 0.723 |
| SEVER. | 0.762 | 0.955 | 0.786 | 0.956 | 0.744 | 0.902 | 0.743 | 0.944 | 0.762 | 0.735 |
| RFC. | 0.800 | 0.969 | 0.805 | 0.966 | 0.812 | 0.945 | 0.785 | 0.963 | 0.783 | 0.961 |

| $(\epsilon = 15\%)$ | Min-Max$(z=0)$ | | Min-Max$(z=1)$ | | F-Attack$(z=0)$ | | F-Attack$(z=1)$ | | F-Attack* | |
|---|---|---|---|---|---|---|---|---|---|---|
| | Acc. | Fair. | Acc. | Fair. | Acc. | Fair. | Acc. | Fair. | Acc. | Fair. |
| No Defense. | 0.802 | 0.945 | 0.800 | 0.945 | 0.759 | 0.946 | 0.800 | 0.672 | 0.789 | 0.721 |
| SEVER. | 0.758 | 0.960 | 0.743 | 0.980 | 0.724 | 0.961 | 0.726 | 0.855 | 0.724 | 0.756 |
| RFC. | 0.805 | 0.955 | 0.801 | 0.967 | 0.775 | 0.953 | 0.801 | 0.955 | 0.774 | 0.945 |

Table 7: RFC & Baseline Methods' Performance on COMPAS dataset for Equalized Treatment

| | No Attack | | Label Flip(10%) | | Label Flip(20%) | | Attr. Flip(10%) | | Attr. Flip(20%) | |
|---|---|---|---|---|---|---|---|---|---|---|
| | Acc. | Fair. | Acc. | Fair. | Acc. | Fair. | Acc. | Fair. | Acc. | Fair |
| No Defense. | 0.671 | 0.845 | 0.663 | 0.854 | 0.655 | 0.845 | 0.659 | 0.873 | 0.670 | 0.845 |
| SEVER. | 0.665 | 0.831 | 0.665 | 0.849 | 0.667 | 0.836 | 0.674 | 0.842 | 0.672 | 0.834 |
| RFC. | 0.671 | 0.830 | 0.669 | 0.845 | 0.672 | 0.846 | 0.677 | 0.850 | 0.672 | 0.844 |

| $(\epsilon = 10\%)$ | Min-Max$(z=0)$ | | Min-Max$(z=1)$ | | F-Attack$(z=0)$ | | F-Attack$(z=1)$ | | F-Attack* | |
|---|---|---|---|---|---|---|---|---|---|---|
| | Acc. | Fair. | Acc. | Fair. | Acc. | Fair. | Acc. | Fair. | Acc. | Fair. |
| No Defense. | 0.680 | 0.843 | 0.662 | 0.845 | 0.666 | 0.862 | 0.658 | 0.854 | 0.677 | 0.843 |
| SEVER. | 0.662 | 0.836 | 0.659 | 0.839 | 0.646 | 0.850 | 0.648 | 0.825 | 0.671 | 0.842 |
| RFC. | 0.677 | 0.838 | 0.676 | 0.841 | 0.675 | 0.852 | 0.674 | 0.848 | 0.668 | 0.845 |

| $(\epsilon = 15\%)$ | Min-Max$(z=0)$ | | Min-Max$(z=1)$ | | F-Attack$(z=0)$ | | F-Attack$(z=1)$ | | F-Attack* | |
|---|---|---|---|---|---|---|---|---|---|---|
| | Acc. | Fair. | Acc. | Fair. | Acc. | Fair. | Acc. | Fair. | Acc. | Fair. |
| No Defense. | 0.629 | 0.866 | 0.668 | 0.845 | 0.669 | 0.820 | 0.662 | 0.855 | 0.675 | 0.855 |
| SEVER. | 0.633 | 0.848 | 0.631 | 0.855 | 0.613 | 0.850 | 0.613 | 0.866 | 0.678 | 0.845 |
| RFC. | 0.659 | 0.845 | 0.674 | 0.846 | 0.675 | 0.835 | 0.671 | 0.846 | 0.680 | 0.845 |

## A.4 GENERALIZING F-ATTACK TO DNN-BASED FAIR LEARNING

In the paper, we have mainly discussed linear models and convex optimization problems. In this part, we argue that our method and conclusions are general and representative, which can easily extend to Deep Neural Networks (DNNs). To show this point, we focus on a representative and popular fair learning method in DNN, called *Learning Fair Representations (LFR)* (Zemel et al., 2013; Edwards & Storkey, 2015; Beutel et al., 2017; Wang et al., 2019). Then, we propose a modified version of F-Attack, called **F-Attack-DNN (FAD)**, which inherits the

Table 8: RFC & Baseline Methods' Performance on COMPAS Dataset using (Zafar et al., 2017).

| | No Attack | | Label Flip(10%) | | Label Flip(20%) | | Attr. Flip(10%) | | Attr. Flip(20%) | |
| --- | --- | --- | --- | --- | --- | --- | --- | --- | --- | --- |
| | Acc. | Fair. | Acc. | Fair. | Acc. | Fair. | Acc. | Fair. | Acc. | Fair |
| No Defense. | 0.651 | 0.844 | 0.642 | 0.865 | 0.644 | 0.827 | 0.651 | 0.855 | 0.662 | 0.840 |
| SEVER. | 0.647 | 0.846 | 0.6770 | 0.841 | 0.657 | 0.830 | 0.644 | 0.851 | 0.642 | 0.830 |
| RFC. | 0.656 | 0.852 | 0.668 | 0.841 | 0.679 | 0.850 | 0.677 | 0.846 | 0.661 | 0.852 |

| $(\epsilon = 10\%)$ | Min-Max($z = 0$) | | Min-Max($z = 1$) | | F-Attack($z = 0$) | | F-Attack($z = 1$) | | F-Attack* | |
| --- | --- | --- | --- | --- | --- | --- | --- | --- | --- | --- |
| | Acc. | Fair. | Acc. | Fair. | Acc. | Fair. | Acc. | Fair. | Acc. | Fair |
| No Defense. | 0.655 | 0.840 | 0.645 | 0.842 | 0.646 | 0.832 | 0.628 | 0.851 | 0.654 | 0.816 |
| SEVER. | 0.645 | 0.860 | 0.640 | 0.861 | 0.628 | 0.840 | 0.619 | 0.852 | 0.659 | 0.831 |
| RFC. | 0.655 | 0.858 | 0.662 | 0.854 | 0.663 | 0.850 | 0.660 | 0.851 | 0.659 | 0.852 |

| $(\epsilon = 15\%)$ | Min-Max($z = 0$) | | Min-Max($z = 1$) | | F-Attack($z = 0$) | | F-Attack($z = 1$) | | F-Attack* | |
| --- | --- | --- | --- | --- | --- | --- | --- | --- | --- | --- |
| | Acc. | Fair. | Acc. | Fair. | Acc. | Fair. | Acc. | Fair. | Acc. | Fair |
| No Defense. | 0.659 | 0.820 | 0.631 | 0.851 | 0.649 | 0.800 | 0.621 | 0.851 | 0.658 | 0.815 |
| SEVER. | 0.641 | 0.866 | 0.638 | 0.851 | 0.596 | 0.865 | 0.611 | 0.870 | 0.643 | 0.831 |
| RFC. | 0.671 | 0.861 | 0.659 | 0.855 | 0.650 | 0.851 | 0.640 | 0.857 | 0.659 | 0.850 |

Table 9: RFC & Baseline Methods' Performance on COMPAS dataset, when $d = 6$

| $(\epsilon = 10\%)$ | Min-Max($z = 0$) | | Min-Max($z = 1$) | | F-Attack($z = 0$) | | F-Attack($z = 1$) | | F-Attack* | |
| --- | --- | --- | --- | --- | --- | --- | --- | --- | --- | --- |
| | Acc. | Fair. | Acc. | Fair. | Acc. | Fair. | Acc. | Fair. | Acc. | Fair. |
| No Defense. | 0.650 | 0.865 | 0.581 | 0.838 | 0.648 | 0.800 | 0.651 | 0.820 | 0.631 | 0.815 |
| SEVER. | 0.668 | 0.853 | 0.643 | 0.788 | 0.652 | 0.800 | 0.656 | 0.802 | 0.644 | 0.823 |
| RFC. | 0.670 | 0.850 | 0.651 | 0.835 | 0.658 | 0.820 | 0.663 | 0.822 | 0.665 | 0.835 |

| $(\epsilon = 15\%)$ | Min-Max($z = 0$) | | Min-Max($z = 1$) | | F-Attack($z = 0$) | | F-Attack($z = 1$) | | F-Attack* | |
| --- | --- | --- | --- | --- | --- | --- | --- | --- | --- | --- |
| | Acc. | Fair. | Acc. | Fair. | Acc. | Fair. | Acc. | Fair. | Acc. | Fair. |
| No Defense. | 0.621 | 0.854 | 0.617 | 0.859 | 0.515 | 0.980 | 0.614 | 0.890 | 0.640 | 0.801 |
| SEVER. | 0.625 | 0.819 | 0.641 | 0.801 | 0.624 | 0.770 | 0.645 | 0.853 | 0.635 | 0.815 |
| RFC. | 0.660 | 0.833 | 0.675 | 0.857 | 0.672 | 0.832 | 0.664 | 0.832 | 0.660 | 0.847 |

idea of F-Attack but accommodate to attack LFR. Finally, we show the experimental results to demonstrate the success of our proposed attack method.

**Learning Fair Representation (LFR).** The goal of learning fair representations is to remove sensitive information from the learned representations, so that the sensitive information will not interfere the model prediction. In this paper, we focus on the method (Edwards & Storkey, 2015) to achieve the goal of LFR. It is an adversarial framework that learns the target classification task without the ability to predict sensitive information. Many fairness learning methods in DNN tasks follow the similar strategy to learn fair representations for images (Wang et al., 2019) and texts (Sarafianos et al., 2019). Figure 3 illustrates the

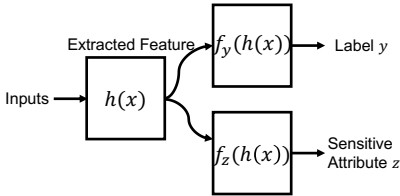

Figure 3: Learning Fair Representation

overview of the framework. In detail, a feature extractor $h(\cdot)$ first extracts latent information form the input. Then, two prediction heads $f_y(\cdot)$ and $f_z(\cdot)$ are attached to the extracted feature space to predict the label $y$ and sensitive attribute $z$ respectively. To ensure there is no sensitive information in the learned representation $h(\cdot)$, the training objective of LFR can be expressed as:

$$\min_{f} \max_{f_s} \mathbb{E}\Big[l\left(f\left(x\right),y\right) - \alpha \cdot l(f_z(h(x)), z)\Big] \qquad (10)$$

Here, the function $f(\cdot) = f_y(h(\cdot))$ is the overall classification model. This objective briefly resembles the idea of GAN (Goodfellow et al., 2020): In the inner maximization, $f_z(\cdot)$ is updated to minimize $l(f_z(h(x)), z)$, which aims to explore helpful information from $h(x)$ to predict the sensitive attribute $z$; In the outer minimization, the model $f(\cdot)$ and feature extractor $h(\cdot)$ are updated to break $f_z(\cdot)$'s ability to predict $z$, while preserve the ability to predict $y$. In this way, the model is trained to eliminate sensitive information from the extracted features.

**F-Attack-DNN (FAD).** Based on the design of LFR, we propose F-Attack-DNN to generate poisoning samples to degrade the performance of LFR. It shares the similar overall optimization objective as in Eq.(3) in Section 3, which solves a min-max problem:

$$\min_{f \in \mathcal{H}_{fair}} \max_{D_P \subseteq \mathcal{F}} L(D_C \cup D_P, w) \tag{11}$$

To solve this problem for DNN-based model $f$, we propose the following algorithm at Algo. 3 called **F-Attack-DNN (FAD)**. Specifically, in the inner maximization, FAD first finds the sample which maximize loss value under current model $f$, which is similar to Eq.(4) in Section 3. Notably, to solve this optimization problem for DNN-based model, we can apply well-known methods such as Projected Gradient Descent (Madry et al., 2017). Then, Step (2) and Step (3) imitate the (re)-training process of LFR to update the model $f$. In detail, Step (2) updates the classifier $f_z$ to predict sensitive attribute $z$ from the learned feature $h(x)$; Step (3) solves the outer minimization for Eq.(11). This will ensure $f$ fall into the space $\mathcal{H}_{fair}$ and $f$ has a minimized training error.

---

**Algorithm 3:** Algorithm of F-Attack-DNN (FAD)

---

**Input** : Clean data $D_C$, number of poisoned samples $\epsilon n$,
feasible set $\mathcal{F}$, $\alpha > 0$, $\eta > 0$, warm-up steps $n_{burn}$.
**Output :** A poisoning set $D_P^*$
**for** $t = 1, ..., n_{burn} + \epsilon n$ **do**

    1. Solve inner maximization: $(x, y, z) = \arg\max_{(x,y,z) \in \mathcal{F}} l(f(x, w), y)$ and let $D_P = \epsilon n \times \{(x, y, z)\}$

    2. Update the discriminator: $f_z = f_z - \eta \cdot \frac{1}{|D_C \cup D_p|} \sum_{(x,y,z) \in D_C \cup D_P} \nabla_{f_z} \Big( l(f_z(h(x)), z) \Big)$

    3. Solve outer minimization: $f = f - \eta \cdot \frac{1}{|D_C \cup D_p|} \cdot \sum_{(x,y,z) \in D_C \cup D_P} \nabla_f \Big( l(f(x), y) - \alpha \cdot l(f_z(h(x)), z) \Big)$

    **if** $t > n_{burn}$ **then**

      | $\quad D_P^* = D_P^* \cup \{(x, y, z)\}$

    **end**

**end**

---

Notably, the only difference between F-Attack (Algo. 1) and F-Attack-DNN (Algo. 3) is about how the outer minimization problem is solved, which is how the fair classification model $f$ is updated. In another word, for various fair learning schemes, we can adaptively modify the objective of F-Attack to accommodate the targeted fair learning methods. Next, we will show experimental results to demonstrate the effectiveness of FAD.

**Experimental Results.** To demonstrate the effectiveness of our proposed attack, we again focus on the Adult Census Dataset and consider the fairness criteria as Equalized True Positive Rate (TPR) between genders. Under this dataset, we consider two layer Multi-Layer Perceptron (MLP) for feature extraction and classification. Then, the sensitive attribute discriminator $f_z(\cdot)$ is attached onto the pen-ultimate layer of MLP. In our experiments, we insert various fraction (from 0% to 40%) of poisoning samples to the training set, and define the feasible injection set $\mathcal{F}$ to be $\{(x, y, z) : ||x - \mu_y|| \leq d\}$, where $d = 9$. In the Table 10, we report the models' goodness of accuracy and fairness, for the models which are obtained via LFR without any defenses. Notably, we choose to report the average model performance (for 10 runs) on the checkpoint when it achieves best validation performance (see Section 4 Eq.9). From the result, we find that FAD can achieve much stronger attacking effectiveness than other baseline attacks, such as label flip, attribute flip and Min-Max Attack (MM) (Steinhardt et al., 2017). Here, MM represents the attack algorithm similar to Algo. 3, but without step 2 and the term about $f_s$ in Step 3. This experimental result implies the generality of F-Attack and its potential to extend to various fairness training schemes. We will leave the study on fair learning for other DNN applications, such as in image domain and text domain for future investigation.

## A.5    THE PRECISION OF THE DETECTION OF RFC

To have a deeper understanding of the defense method RFC, we check the calculated poisoning scores for all training samples in a poisoned Adult Census Dataset. In Figure 4, we visualize the distribution of poisoning scores for clean samples, as well as poisoning samples from 4 versions of F-Attack. Specifically, we report the scores for each subgroup separately and we use red dots to denote the poisoning samples. From this figure, it is easy to see that the poisoning samples always appear at the tail in their labeled subgroup with (almost) lowest poisoning scores. Note that during our proposed defense method RFC, at each iteration, it only removes a small

Table 10: *F-Attack-DNN (FAD)* vs. Baseline Attacks on Adult Census Dataset

| | Label. Flip | | Attr. Flip | | MM (Z = 0) | | MM (Z = 1) | | FAD (Z = 0) | | FAD (Z = 1) | |
|---|---|---|---|---|---|---|---|---|---|---|---|---|
| Ratio. | Acc. | Fair. | Acc. | Fair. | Acc. | Fair. | Acc. | Fair. | Acc. | Fair. | Acc. | Fair. |
| 0% | 0.797 | 0.960 | 0.797 | 0.960 | 0.797 | 0.960 | 0.797 | 0.960 | 0.797 | 0.960 | 0.797 | 0.960 |
| 10% | 0.781 | 0.972 | 0.795 | 0.960 | 0.774 | 0.970 | 0.778 | 0.960 | **0.757** | **0.936** | 0.775 | 0.954 |
| 20% | 0.783 | 0.972 | 0.798 | 0.958 | 0.776 | 0.960 | 0.740 | 0.963 | 0.731 | **0.945** | **0.713** | 0.952 |
| 30% | 0.766 | 0.967 | 0.795 | 0.958 | 0.724 | 0.977 | 0.692 | 0.967 | 0.676 | **0.947** | **0.644** | 0.965 |
| 40% | 0.741 | 0.971 | 0.797 | 0.936 | 0.701 | 0.971 | 0.668 | 0.987 | 0.655 | **0.924** | **0.647** | 0.961 |

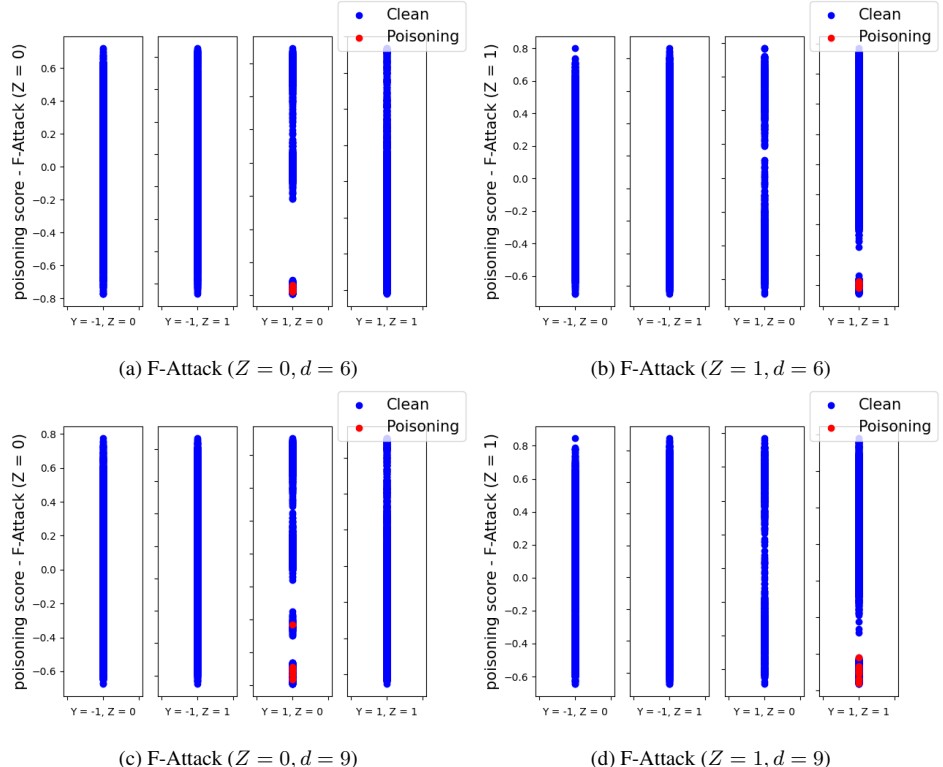

(a) F-Attack ($Z = 0, d = 6$)        (b) F-Attack ($Z = 1, d = 6$)

(c) F-Attack ($Z = 0, d = 9$)        (d) F-Attack ($Z = 1, d = 9$)

Figure 4: The Poisoning Score calculated by RFC for samples from each subgroup.

portion of samples from a particular subgroup with highest or lowest poisoning score. As a result, RFC will not remove too many clean samples until they can find the poisoning samples. Besides, one interesting fact is: under this Adult Census Dataset, F-Attack prefers to add poisoning samples with $y = 1$.

