# OpenReview forum: "Towards Fair Classification against Poisoning Attacks"
_ICLR.cc/2023/Conference — Submitted to ICLR 2023_

### Official Review · Reviewer_Qsnj · 2022-10-24

**Confidence:** 5
**Clarity, Quality, Novelty And Reproducibility:** I am a bit concerned about the novelt…
**Correctness:** 4
**Technical Novelty And Significance:** 3
**Empirical Novelty And Significance:** 3
**Recommendation:** 5

**Strength And Weaknesses:**

Strength:

(1). The paper performed extensive experiments and detailed analysis on the experimental results, which clearly conveyed two critical points: i) fair classification learners, although equipped with existing defense strategies such as k-NN and Sever, are not robust enough against data poisoning attacks. ii) the RFC defense mechanism proposed in this paper indeed helps mitigate the effect of data poisoning attacks on fair learners and outperforms existing defense methods.

(2). The paper is easy to read. I can quickly understand the topic and the key research points after reading the problem statements. The presentation and paper structure is also nice and clear.

Weaknesses:

(1). The paper is technically weak from a scientific point of view. While the paper performed extensive experiments and also designed interesting attack and defense algorithms, I find it hard to justify that their approaches contain very novel techniques compared to prior works. Their attack algorithm is pretty straightforward and looks like adapted from existing research outcomes. The minimax formulation also applies to traditional classification victim learners. The only difference here is that in this paper, the min operator is restricted to models that satisfy the fairness constraint. Therefore, I don't see why the attack algorithm is technically novel. Similarly, as the authors pointed out, the defense here also borrows some ideas from prior works. Overall, I am a bit concerned about not having enough novelty and technical contribution for this paper.

(2). The theoretical results do not appear appealing to me because it does not state directly to the point that how much improvement in the accuracy or fairness is guaranteed by the defense. Right now, the results are somewhat weak because it's about the poisoning score. In contrast, the more interesting theoretical problem is how much accuracy is preserved when using the RFC defense compared to without using RFC. I am wondering if the theoretical part of this paper can be improved in that way.

(3). In the objective approximation equation (i.e., the equation between (2) and (3)), the last equality does not make sense to me. Instead of an equality, I believe that should be an inequality. I am wondering if that's a typo or did I miss anything?

(4). The scope of this paper is limited to classification only, Furthermore, it's restricted to convex loss functions. As a result, the results in this paper are not directly applicable to DNN-based classification learners. I believe the paper can easily generalize their results, e.g., by designing different poisoning algorithms for DNN-based learners. The minimax formulation no longer works but from a practical perspective, I don't really think the minimax formulation is super critical to this paper. There are a variety of poisoning attacks against DNN, and these attacks can be easily adapted to the fair classification setting. Please correct me if I am wrong.

**Summary Of The Paper:**

This paper studied data poisoning attacks against fair classification algorithms. The objective of poisoning a fair learner is approximated by a minimax optimization problem. When the loss function is convex, one can switch the min-max operator and use gradient descent to construct the optimal poisoning dataset. On the basic fair learner and a few robust variations such as k-NN based defense and Sever-based defense algorithms, the authors demonstrate that their poisoning algorithm can reduce the accuracy and the fairness metric of the victim learners. Motivated by that, the paper designed a defense mechanism called RFC, which employs similar ideas as prior works, but adapted to the fair classification setting. The paper analyzed the relationship between the poisoning scores of clean and poisoned samples. Extensive experiments on abundant datasets show that the proposed defense can enhance the resilience of fair learner against poisoning attacks.

**Summary Of The Review:**

I worked in related areas.

---

> ### Author Response · Authors · 2022-11-20
> **Response to reviewer Qsnj (part 1)**
>
>
> Thank you for reviewing our paper and we are happy to read that the paper is well written. Based on the suggestions of the reviewer, we will provide more clarifications on: (1) The contribution of our proposed F-Attack. (2) Can the idea of F-Attack be generalized to DNN-based models? (3) The contribution of our proposed defense RFC. (4) The theoretical result. (5) Some detailed issues.
>
> **Q1. Contribution of the attack algorithm.**
>
> The reviewer questions the technical contribution of our attack method F-Attack, by mentioning F-Attack is an extension of previous method which only introduces fairness constraints in the bi-level optimization process. However, we would politely argue that some key merits of our attack are overlooked, as:
> 1. "Simplicity'' is an advantage, instead of a weakness. We agree with the reviewer that "the only difference of our attack and [1] is that, the min operator in our attack is restricted to models that satisfy the fairness constraint''. However, our study shows that **only this change is sufficiently effective to greatly degrade the fair classifiers' performance**. This analysis can well support our claim: fair classification \& existing defenses can be easily broken by poisoning attacks such as F-Attack.
> 2. More importantly, our method is general and representative. As the reviewer mentioned,  "the only difference of our attack and previous method is that, the min operator is restricted to models that satisfy the fairness constraint''. Therefore, one can easily accommodate the idea of F-Attack to other fair learning scenarios or other type of poisoning attacks, by simply considering ``the (re)-trained model satisfy the fairness constraint''. In the following part, we will use an extra experimental study to illustrate how the idea of F-Attack can be easily generalized to other fair learning scenarios.

---

> > ### Author Response · Authors · 2022-11-20
> > **Response to reviewer Qsnj (part 2)**
> >
> > **Q2. Can the attack be generalized to DNN-based models?**
> >
> > As discussed in the answer to Q1, the core idea of F-Attack is to consider “the (re)-trained model satisfy the fairness constraint”. We believe this core idea can be easily accommodate to attack DNN-based fair learning. To demonstrate this point, in Appendix A.4 in our revised paper, we provide an extra empirical study in the setting of DNN-based fair learning.
> >
> > In detail, we focus on one representative fair learning method in DNN to learn fair representations [2]. Basically, it is an adversarial framework that learns the target classification task, but also remove the sensitive information from the learned representations.
> > To poison this learning algorithm, we propose **F-Attack-DNN (FAD)** which is a simple generalization of F-Attack. In FAD, it also solves similar overall optimization objective as in Eq.(3) in Section 3. The only difference is: in the outer-minimization, FAD imitates the (re)-training process of fair training [2] to update the model.
> > In fact, the only difference between FAD and F-Attack is about how the outer-minimization problem is solved, which is how the fair classification model  is updated during the (re)-training process. Plz refer Appendix A.4 for more detailed discussion about the fair method we studied and our proposed FAD Attack. (Notably, we agree with the reviewer that in DNN-based model, the assumption of min-max formulation does not hold anymore because the functions are not convex. However, we found the min-max algorithm works well in practice for our studied MLP model.)
> >
> > To demonstrate the effectiveness of our proposed FAD Attack, we again focus on the Adult
> > Census Dataset and consider the fairness criteria as Equalized True Positive Rate (TPR) between genders. Under this dataset, we consider two layer Multi-Layer Perceptron (MLP) for feature extraction and classification. In our experiments, we
> > insert various fraction (from 0\% to 40\%) of poisoning samples to the training set. In the following table, we report the models’ goodness of accuracy and fairness (for 10 runs), for the models which are obtained via [2] without any defenses. From the result, we find that FAD can achieve much stronger attacking effectiveness than other baseline attacks, such as label flip (FL), attribute flip (AF) and Min-Max Attack (MM) with Z = 0 or 1. This experimental result implies the generality of F-Attack and its potential to extend to various fairness training schemes.
> >
> > |              | LF    |       | AF    |       | MM0   |       | MM1   |       | FAD0      |           | FAD1      |        |
> > |--------------|-------|-------|-------|-------|-------|-------|-------|-------|-----------|-----------|-----------|--------|
> > | Ratio        | Acc   | Fair  | Acc   | Fair  | Acc   | Fair  | Acc   | Fair  | Acc       | Fair      | Acc       | Fair   |
> > | 0%           | 0.797 | 0.960 | 0.797 | 0.960 | 0.797 | 0.960 | 0.797 | 0.960 | 0.797     | 0.960     | 0.797     | 0.960  |
> > | 10%          | 0.781 | 0.972 | 0.795 | 0.960 | 0.774 | 0.970 | 0.778 | 0.960 | **0.757** | **0.936** | 0.775     | 0.954 |
> > | 20%          | 0.783 | 0.972 | 0.798 | 0.958 | 0.776 | 0.960 | 0.740 | 0.963 | 0.731     | **0.945** | **0.713** | 0.952  |
> > | 30%          | 0.766 | 0.967 | 0.795 | 0.958 | 0.724 | 0.977 | 0.692 | 0.967 | 0.6767    | **0.947** | **0.644** | 0.965  |
> > | 40%          | 0.741 | 0.971 | 0.797 | 0.936 | 0.701 | 0.971 | 0.668 | 0.987 | 0.655     | **0.924** | **0.647** | 0.961  |

---

> > > ### Author Response · Authors · 2022-11-20
> > > **Response to reviewer Qsnj (part 3)**
> > >
> > >
> > > **Q3. Contribution of the proposed defense RFC.**
> > >
> > > We agree with the reviewer that our proposed algorithm in Algo.(2) in the paper also has some similarities with SEVER~[3]. For example, both our method RFC and [3] utilize Singular Value Decomposition (SVD) to scout abnormal training samples. However, our proposed defense RFC also have particular significance as:
> > > * We propose a novel "Online Sanitization Process'' which is critical in the setting of fair classification. In our method RFC, we scout abnormal samples from each individual sensitive subgroup in each class (in each iteration). Then, we only remove the samples from the subgroup whose removal can result the most performance ``recovery''. This process is critical for fair classification. It is because: in fair classification problems, it is usual that some subgroups from certain classes have significantly fewer samples than other subgroups.
> > >  Previous methods, such as SEVER [3] which is innocent to this condition, are very likely to remove clean samples from minority subgroups, which can greatly hurt the model performance.
> > > * Furthermore, similar to our proposed F-Attack, RFC is also general and possible to be extended to various model architectures, such as Deep Neural Networks (DNNs), for robust fair classification. For example, we can apply the Energy-based Out-of Distribution Detection Liu et al [4] to find the abnormal samples from each subgroup. Then, we follow a similar manner as RFC to propose poisoning sets and conduct fair classification.
> > >
> > >
> > > **Q4. More clarification on theoretical results**
> > >
> > > We thank the reviewer's suggestion to theoretically analyze ``how much accuracy is preserved with using RFC''. Due to the time limit and the relative complexity for the derivation (because RFC involves an Online Data Sanitization process), we will leave this analysis for future investigation. However, we would like to provide extra evidence which further demonstrates the effectiveness of RFC.
> > >
> > > In our revised paper in Appendix A.5, we check the calculated poisoning scores for all
> > > training samples in a poisoned Adult Census Dataset. In Figure 4, we visualize the distribution of poisoning scores for clean samples, as well as poisoning samples from 4 versions of F-Attack. From the figure, it is easy to see that the poisoning samples always appear at the tail in their labeled subgroup with (almost) lowest poisoning scores. Note that during our proposed defense method RFC, at each iteration, it only removes a small portion of samples from a particular subgroup with highest or lowest poisoning score. As a result, RFC will not remove too many clean samples until they can find the poisoning samples. Therefore, RFC can effectively preserve the accuracy of fair classification.
> > >
> > >
> > >
> > > **Q5. Should there be an $\leq$ instead of $=$, between Eq.(2) and Eq.(3) in our paper?**
> > >
> > > We apologize for the possible unclearness. In Section 2 of our paper, we mention ``we define the empirical loss function as $L(D, w)$ as the average loss value of the model'', which is the sum of training loss divided by the number of training samples. Given that the sets $D_C$ and $D_P$ have the size $n$ and $\epsilon \cdot n$, we have the following calculations (we use $l(x_i)$ to represent the model loss on a specific sample $x_i$):
> > >
> > > (1) $L(D_C, w) = \frac{1}{n} \sum_{i \in D_C} l(x_i)$
> > >
> > > (2) $\epsilon \cdot L(D_P, w) = \epsilon \cdot  \frac{1}{\epsilon \cdot n} \sum_{i \in D_P} l(x_i) = \frac{1}{n} \sum_{i \in D_P} l(x_i)$
> > >
> > > Combine Eq.(1) and Eq.(2)
> > >
> > > $L(D_C, w)+ \epsilon \cdot L(D_P, w) = \frac{1}{n} \sum_{i \in D_C \cup D_P} l(x_i) = (1+\epsilon) L(D_C \cup D_P)$
> > >
> > >
> > >
> > > **References**
> > >
> > > [1] Certified Defenses for Data Poisoning Attacks, Steinhardt et al, NeurIPS 2017
> > >
> > > [2] Censoring representations with an adversary. Edwards \& Storkey et al, ICLR 2016
> > >
> > > [3] Sever: A Robust Meta-Algorithm for Stochastic Optimization, Diakonikolas et al, ICML 2019
> > >
> > > [4] Energy-based out-of-distribution detection, Liu et al, NeurIPS 2020

---

### Official Review · Reviewer_MXgE · 2022-10-25

**Confidence:** 4
**Correctness:** 3
**Technical Novelty And Significance:** 2
**Empirical Novelty And Significance:** 2
**Recommendation:** 5

**Clarity, Quality, Novelty And Reproducibility:**

The paper is mostly well-written and is easy to follow. However, some of the statements in the paper are inaccurate. For example, upper bound (ii) may not be tight even when $\epsilon$ is small, as it is related to the property of the training loss and also the search space of the poisoning points. In extreme cases, the poisoned loss can be much higher than the average clean loss and hence the upper bound can be very loose. In Eq. (3), the authors should provide more details on why we choose to solve the "min-max" form when the loss function is convex. Is it of similar reason as in Steinhardt et al., (2017)? Specifically, the (easier to solve) "min-max" form is a universal upper bound to the original "max-min" form and also has an asymptotic convergence when the loss function is convex. As for the originality, the core techniques for the attacks and defenses are mostly from Steinhardt et al., (2017) and  Diakonikolas et al., (2019). Therefore, I am worried that originality is somewhat limited.

**Strength And Weaknesses:**

Strength:
1. vulnerability of fair classification under data poisoning is interesting.
2. the proposed attack and defense performs better than the existing attacks and defenses.

Weakness:
1. the major concern is in the significance of this work: the F-attack is a variant of the existing min-max attack, where the major difference is how the model weight is updated after the poisoning point is selected. The proposed defense is also a variant of the Sever defense, with finer analysis of the training data at particular subgroups. Therefore, the technical novelty of the paper is limited.
2. the considered attacks are quite weak and may not even be a real threat. The fraction of poisoning points is >10%, which is considered quite a high ratio in practical applications. However, even with such a high poisoning ratio, the attack effectiveness is limited. Taking the poisoned model's test accuracy as an example, we expect a strong poisoning attack to induce much higher test error in comparison to the poisoning ratio $\epsilon$, while results in this paper induce test error smaller than $\epsilon$. Therefore, one may conclude that current fair classifications are not under severe threat from poisoning attacks.
3. As for the evaluation, one relevant baseline is missing (Jo et al., 2022). The min-max attack may also be improved by leveraging target models, which is done in Koh et al., (2021) and also demonstrated similarly in Suya et al., (2021).

Jo et al., "Breaking Fair Binary Classification with Optimal Flipping Attacks", ISIT 2022.
Suya et al., "Model-Targeted Poisoning Attacks with Provable Convergence", ICML 2021.

**Summary Of The Paper:**

This paper studies data poisoning attacks against fair classification algorithms, and proposes a new poisoning attack, which is a variant of the existing min-max attack. In addition, a robust fair classification defense is proposed, which is an extension of the Sever defense into fair classification problems. Empirical results show that the proposed attack is more effective than the existing min-max attack against different baseline defenses and the proposed defense is also robust against the newly proposed attack.

**Summary Of The Review:**

The idea of studying fair classification under poisoning attacks is interesting. However, the core techniques in this paper are mostly based on existing poisoning attacks and defenses for traditional classification and hence, have limited technical novelty. In addition, the proposed attacks also have limited effectiveness empirically, which further weakens the results. Considering all these factors, a weak reject is recommended.

---

> ### Author Response · Authors · 2022-11-20
> **Response to reviewer MXgE (part 1)**
>
> Thank you for reviewing our paper and we are happy to read that our studied problem is interesting and the paper is well written. Based on the suggestions of the reviewer, we will provide more clarifications on: (1) The contribution of our proposed F-Attack. (2) The contribution of our proposed defense RFC. (3) Is the attack too weak? (4) Comparison with baseline method.   (5) Some detailed issues.
>
> **Q1. Contribution of the attack algorithm.**
>
> The reviewer mentions our proposed F-Attack is an extension of previous Min-Max attack method [1] into fair classification. However, we would politely argue that some key merits of our attack are overlooked, as:
>  * "Simplicity'' is an advantage, instead of a weakness. We agree with the reviewer that ``the only difference of our attack and [1] is that, the min operator in our attack is restricted to models that satisfy the fairness constraint''. However, our study shows that **only this change is sufficiently effective to greatly degrade the fair classifiers' performance**. This analysis can well support our claim: fair classification \& existing defenses can be easily broken by poisoning attacks such as F-Attack.
> * More importantly, our method is general and representative. As the reviewer mentioned,  "the only difference of our attack and previous method is that, fairness constraint is considered''. Therefore, one can easily accommodate the idea of F-Attack to other fair learning scenarios or other type of poisoning attacks, by simply considering ``the (re)-trained model satisfy the fairness constraint''. In the following part, we will use an extra experimental study to illustrate how the idea of F-Attack can be easily generalized to other fair learning scenarios.

---

> > ### Author Response · Authors · 2022-11-20
> > **Response to reviewer MXgE (part 2)**
> >
> > **F-Attack is general and representative.**
> >
> > Given that the core idea of F-Attack is to consider “the (re)-trained model satisfy the fairness constraint”. We believe this core idea can be easily accommodate to attack DNN-based fair learning. To demonstrate this point, in Appendix A.4 in our revised paper, we provide an extra empirical study in the setting of DNN-based fair learning.
> >
> > In detail, we focus on one representative fair learning method in DNN to learn fair representations [2]. Basically, it is an adversarial framework that learns the target classification task, but also remove the sensitive information from the learned representations.
> > To poison this learning algorithm, we propose **F-Attack-DNN (FAD)** which is a simple generalization of F-Attack. In FAD, it also solves similar overall optimization objective as in Eq.(3) in Section 3. The only difference is: in the outer-minimization, FAD imitates the (re)-training process of fair training [2] to update the model.
> > In fact, the only difference between FAD and F-Attack is about how the outer-minimization problem is solved, which is how the fair classification model  is updated during the (re)-training process. Plz refer Appendix A.4 for more detailed discussion about the fair method we studied and our proposed FAD Attack.
> >
> > To demonstrate the effectiveness of our proposed FAD Attack, we again focus on the Adult
> > Census Dataset and consider the fairness criteria as Equalized True Positive Rate (TPR) between genders. Under this dataset, we consider two layer Multi-Layer Perceptron (MLP) for feature extraction and classification. In our experiments, we
> > insert various fraction (from 0\% to 40\%) of poisoning samples to the training set. In the following table, we report the models’ goodness of accuracy and fairness (for 10 runs), for the models which are obtained via [2] without any defenses. From the result, we find that FAD can achieve much stronger attacking effectiveness than other baseline attacks, such as label flip (FL), attribute flip (AF) and Min-Max Attack (MM) with Z = 0 or 1. This experimental result implies the generality of F-Attack and its potential to extend to various fairness training schemes.
> >
> > |              | LF    |       | AF    |       | MM0   |       | MM1   |       | FAD0      |           | FAD1      |        |
> > |--------------|-------|-------|-------|-------|-------|-------|-------|-------|-----------|-----------|-----------|--------|
> > | Ratio        | Acc   | Fair  | Acc   | Fair  | Acc   | Fair  | Acc   | Fair  | Acc       | Fair      | Acc       | Fair   |
> > | 0%           | 0.797 | 0.960 | 0.797 | 0.960 | 0.797 | 0.960 | 0.797 | 0.960 | 0.797     | 0.960     | 0.797     | 0.960  |
> > | 10%          | 0.781 | 0.972 | 0.795 | 0.960 | 0.774 | 0.970 | 0.778 | 0.960 | **0.757** | **0.936** | 0.775     | 0.954 |
> > | 20%          | 0.783 | 0.972 | 0.798 | 0.958 | 0.776 | 0.960 | 0.740 | 0.963 | 0.731     | **0.945** | **0.713** | 0.952  |
> > | 30%          | 0.766 | 0.967 | 0.795 | 0.958 | 0.724 | 0.977 | 0.692 | 0.967 | 0.6767    | **0.947** | **0.644** | 0.965  |
> > | 40%          | 0.741 | 0.971 | 0.797 | 0.936 | 0.701 | 0.971 | 0.668 | 0.987 | 0.655     | **0.924** | **0.647** | 0.961  |
> >
> > **Q2. The contribution of our proposed defense RFC**
> >
> > We agree with the reviewer our proposed algorithm in Algo.(2) in the paper has some similarities with SEVER~[3]. For example, both our method RFC and SEVER [3] utilize Singular Value Decomposition (SVD) to scout abnormal training samples. However, our proposed defense RFC also have particular significance as:
> > * We propose a novel "Online Sanitization Process'' which is critical in the setting of fair classification. In our method RFC, we scout abnormal samples from each individual sensitive subgroup in each class (in each iteration). Then, we only remove the samples from the subgroup whose removal can result the most performance ``recovery''. This process is critical for fair classification. It is because: in fair classification problems, it is usual that some subgroups from certain classes have significantly fewer samples than other subgroups.
> >  Previous methods, such as SEVER [3] which is innocent to this condition, are very likely to remove clean samples from minority subgroups, which can greatly hurt the model performance.
> > * Furthermore, similar to our proposed F-Attack, RFC is also general and possible to be extended to various model architectures, such as Deep Neural Networks (DNNs), for robust fair classification. For example, we can apply the Energy-based Out-of Distribution Detection Liu et al [4] to find the abnormal samples from each subgroup. Then, we follow a similar manner as RFC to propose poisoning sets and conduct fair classification.

---

> > > ### Author Response · Authors · 2022-11-20
> > > **Response to reviewer MXgE (part 3)**
> > >
> > > **Q3. Is the attack weak?**
> > >
> > > The reviewer mentions that our F-Attack is "weak'',  However, we would disagree with the reviewer from two aspects.
> > > 1. In Table 1, we only consider the model has already be protected by existed strong defenses (for traditional classification), such as SEVER and kNN. If there is no defense, (check the "No Defense'' setting in Table 1 \& Table 2), the model can greatly suffer from the attack. For example, when inserted 10\% poisoning samples, the trained model has a fairness around 0.8, which is much smaller than the fair classification model obtained from clean set. Moreover, if there is no defense, the attacker can choose a larger "feasible injection set'' (Section 3.2), to allow larger searching space of poisoning samples.
> > > 2. In our paper, we consider datasets such as Adult Census and COMPAS. They have the input data with relatively low dimensions, which are 20 and 11 respectively. Therefore, in the low dimension scenario, it could be hard for poisoning attacks to degrade the model performance. This fact is also claimed by the paper [1]. Thus, to further demonstrate that our attack is strong in high-dimensional scenario, we further conduct an experiment of F-Attack against SEVER defense on IMDb review dataset.
> > >
> > > |       | FA0   |       | FA1   |       |
> > > |-------|-------|-------|-------|-------|
> > > | Ratio | Acc   | Fair  | Acc   | Fair  |
> > > | 0%    | 0.826 | 0.922 | 0.826 | 0.922 |
> > > | 3%    | 0.759 | 0.901 | 0.755 | 0.891 |
> > > | 5%    | 0.709 | 0.906 | 0.700 | 0.903 |
> > >
> > > Notably, in IMDb dataset, the data inputs are the movie reviews that represented by bag-of-word embedding (which is 1000 dimension). Then, we focus on a subset that contains data from two genre categories (“Drama”, “Action”) as the sensitive attribute and we desire the model to satisfy Equalized TPR among the sensitive groups. From the result, we can see that both F-Attack (z = 0) and F-Attack (z = 1) (FA0 \& FA1) can result severe accuracy degradation by only inserting 3\% and 5\% poisoning samples, even under the protection of SEVER defense.
> > >
> > >
> > > **Q4. Compare with more baselines?**
> > >
> > > Thanks for mentioning the paper Jo et al 2022. However, we found this paper only aims to degrade the model fairness. For example, in the only empirical study in Jo et al (based on a rather simple simulated dataset), their method can degrade the fairness but also result accuracy improvement. However, out proposed F-Attack can hurt both model accuracy and fairness. For most cases, F-Attack can degrade them simultaneously.
> > >
> > > For the other papers Koh 2018 and Suya et al 2021, they study the problem where the attacker aims to induce a victim model as close as possible to that target model, which is defined as "model-targeted attacks'' in the paper Suya et al [6]. However, our attack is "objective-driven'' attack where the attacker aims to degrade model performance. Moreover, our paper considers to degrade both model accuracy or fairness, which is not covered by these studies.
> > >
> > > **Q5. More detailed issues in "Clarity, Quality, Novelty And Reproducibility''**
> > >
> > > (1) About the tightness of upper bound (ii): we agree with the reviewer that the poisoning samples can have high loss. However, in our paper, we also constrain the poisoning samples to fall into the "feasible injection set'' to limit the space of poisoning samples. For example, see Section 3.2, we constrain the poisoning samples to not be too far from the class center. In this case, we can ensure the loss of poisoning samples is not too much larger than clean samples. (2) Why choose min-max form? We agree with the reviewer that the reason is similar to [1], which is that this form is easier to solve.
> > >
> > >
> > >
> > >
> > > **References**
> > >
> > > [1] Certified Defenses for Data Poisoning Attacks, Steinhardt et al, NeurIPS 2017
> > >
> > > [2] Censoring representations with an adversary. Edwards \& Storkey et al, ICLR 2016
> > >
> > > [3] Sever: A Robust Meta-Algorithm for Stochastic Optimization, Diakonikolas et al, ICML 2019
> > >
> > > [4] Energy-based out-of-distribution detection, Liu et al, NeurIPS 2020
> > >
> > > [5] Breaking Fair Binary Classification with Optimal Flipping Attacks, Jo et al, ISIT 2022.
> > >
> > > [6] Model-Targeted Poisoning Attacks with Provable Convergence. Suya et al, 2021

---

> > > > ### Comment · Reviewer_MXgE · 2022-11-28
> > > > **Response to author rebuttal**
> > > >
> > > > Thanks for the response, but my concerns still remain.
> > > > 1. Results on IMDB: F-Attack is impressive for this case, but I am curious about the performance of Min-Max attack in this case. I agree with the authors that Adult dataset might itself be quite robust to poisoning. I am worried that, the vulnerability is mainly attributed to the dataset itself, not with particular attack strategies.
> > > > 2. More baselines: I am not sure why only violating the fairness constraint is considered a weakness. If the main claim in the paper is about how fair classification is vulnerable under poisoning, then it naturally makes sense to consider violating fairness constraint as much as possible. Also, if the baseline attack can degrade the fairness constraint much more than F-Attack, without impacting the model accuracy, how does these two methods compare? I also believe Jo et al., (2022) mainly violates the fairness constraint mainly because the target models are generated to be only violating fairness constraint. If the target models are also given to violate both the accuracy and fairness, then the method still be doable. As for Koh et al., (2021) and Suya et al., (2021), my main point is, the performance of baseline min-max attack (in terms of reducing the overall test error) can be improved using ambitious (e.g., models with high test error) target models, and this is demonstrated in both two papers. I am just suggesting the authors some possible ways to obtain stronger baseline results. Now I realized that the F-Attack itself may also benefit from target models, because, it is quite similar to the min-max attack.
> > > > 3. Another suggestion when presenting the results is, can the authors report the accuracy of the model based on the specific groups, instead of the general accuracy? I believe this way, the results will be more meaningful.

---

> > ### Comment · Reviewer_MXgE · 2022-11-28
> > **Concerns still remain**
> >
> > I appreciate the authors for addressing some of my concerns. However, my concerns still remain:
> > 1. "Simplicity" is an advantage: I agree that "simple" and "novel" method should be appreciated. However, if a method is mostly "plug-and-use", then the novelty will be limited. To be more specific, for the F-Attack, if the inner and outer optimization part requires special treatment when dealing with the fairness constraint, then this will be considered novel. But (at least from what is presented in the paper), both operations are pretty standard and that is why I am concerned about the significance of this paper.
> > 2. F-Attack is indeed general and representative, although for non-convex models, the convergence properties (in terms of the total training loss) no longer holds. But how much is this generality comes from the design of F-attack itself? The original min-max attack is also compatible to any other types of classifiers because what it needs is to provide the model weight after retraining.

---

### Official Review · Reviewer_Bdkz · 2022-10-30

**Confidence:** 3
**Clarity, Quality, Novelty And Reproducibility:** The work was clear enough to read, an…
**Correctness:** 4
**Technical Novelty And Significance:** 3
**Empirical Novelty And Significance:** 4
**Recommendation:** 5

**Strength And Weaknesses:**

## Strengths:
1. The paper is well rounded and thorough - the authors describe the setting and the problem well, introduce an attack, and then describe a defense to this attack.
2. The results, at least in the setting described in the paper (linear classifier, adult census dataset) improve upon existing techniques.

## Weaknesses:
1. Significance - I'm not convinced this work is significant enough to warrant publication at a top tier conference. While fair classification is of great interest, and data poisoning attacks at well, a defense against data poisoning for fair classification networks seems a bit too niche/lacking general interest.

2. The attack seems like a straightforward extension of the technique in [1] with a straightforward addition of penalty to encourage fairness in the inner optimization problem for poisoning. Granted, the proposed defense does seem novel.

3. The setting for this paper seems weak. The poisoning attacks allow for techniques like label flips, and it seems like a simple linear network is used for the victim model (avoiding any questions of white-box vs. black-box attacks). This is in contrast to recent, more sophisticated poisoning attacks that are in "clean-label" setting, and/or against deep networks.


### Random tidbits:
- I believe the equation in between eqns 2, 3 should have an $\leq$, and not $=$ in the last part.


[1] Muñoz-González, Luis, et al. "Towards poisoning of deep learning algorithms with back-gradient optimization." Proceedings of the 10th ACM workshop on artificial intelligence and security. 2017.


**Summary Of The Paper:**

The authors first describe an attack to poison fair classifiers, and then introduce a defense (filtering based) to mitigate poisoning attacks on fair classifiers.

**Summary Of The Review:**

While I think the paper is well presented, and thorough, the combination of the attack scenario being somewhat niche, and the attack constraints being rudimentary, I don't think the work is of sufficient interest at this point to warrant publication at a top conference.

I am, however, willing to change my score in response to new information/reasoning from the authors.

---

> ### Author Response · Authors · 2022-11-20
> **Response to reviewer Bdkz (part 1)**
>
>
> Thank you for reviewing our paper and we are happy to read that our paper is well rounded and thorough. In the following, we will clarify your key concerns. Based on the suggestions of the reviewer, we will discuss: (1) Why our studied problem is important? (2) Is our attack a simple extension of Muñoz-González? (3) Is the setting of our paper weak? (4) More clarifications on the details of the paper. Next, we will answer to each of these questions.
>
> **Q1. Is the problem about poisoning attacks in fair classification important?**
>
> Our studied problem about ``poisoning attacks in fair ML'' actually follows the studies [1,2,3,4,5] with a long history, which consider the data corruption / adversarial perturbation problems of fair classification.
> In fact, for many applications of fair ML models, such as healthcare, finance or resume screening, a trained model must not discriminate customers based on sensitive attributes including age, sex, or religion. However, most of these fair ML applications often rely on training datasets which are collected from external resources or customers themselves. Therefore, there could be various source of noise in the collected dataset, the model training must be resilient against noisy, subjective, or even adversarial data. For example,
> * The papers [1,2] studied the problem of noisy labels in the training dataset of fair ML problems.
> * The papers [3,4,5] studied the problem of noisy attribute of fair classification. For example, respondents to a survey may choose to conceal or obfuscate their group identity to avoid potential discrimination.
>
> Therefore, for fair classification problems, it is essential to consider corruptions or adversarial perturbations in the training dataset. Compared with previous works, our paper considers the attacker has the capacity to insert arbitrary prediction features, sensitive attribute and with arbitrary labels. It can be seen as a stronger attack than previous studied problems. In our experiment, we also demonstrate the advantage of our defense over previous methods such as [1, 3].
>
>
> **Q2. Is our attack a simple extension of Muñoz-González [6]?**
>
> We appreciate the reviewer for mentioning the paper [6], which is also relevant to our study. We find the paper [6] has a similar adversarial objective as our paper but a very different algorithm to solve the problem.
> Next, we are going to discuss the major similarity and dissimilarity between our paper and paper [6] in detail.
>
> **The similarity.** Our paper has the similar adversarial goal with the paper [6]. In detail, both attacks in our paper and [6] has the objective: to find a poisoning set $D_P$ inserted to the clean dataset $D_C$, if the trained model has a small empirical loss on the poisoned dataset $D_C \cup D_P$, it will has a large test loss. Formally, both objectives can be formed as: (Plz refer Eq.(2) and (3) in [6] and Eq.(2) in our paper)
>
> $\max_{D_P}  E [l(f^*(x), y)]$   s.t.  $f^* = arg\min_f L(D_C \cup D_P)$    (1)
>
>
> **The dissimilarity.** However, the algorithm of our paper to solve the objective is very different from [6]. In detail, [6] applies the **back-gradient** optimization method, where the back-gradient strategy is leveraged to estimate the gradient of the outer objective, in order to guide the update of the poisoning dataset $D_P$. However, our proposed attack method F-Attack transforms the major objective in (1) into a min-max problem. In the scenario of linear classification, our method has guaranteed convergence and close to optimal attacking effectiveness [7].

---

> > ### Author Response · Authors · 2022-11-20
> > **Response to reviewer Bdkz (part 2)**
> >
> > **Q3. Is the setting of our paper weak?**
> >
> > We admit that we only study linear model in the main paper, which is relatively simply but insightful. Moreover, we would also argue that **our attack is general and representative, because it can be easily accommodate to attack DNN-based fair learning**. To demonstrate this point, in Appendix A.4 in our revised paper, we provide an extra empirical study in the setting of DNN-based fair learning.
> >
> > In detail, we focus on one representative fair learning method in DNN to learn fair representations [8]. Basically, it is an adversarial framework that learns the target classification task, but also remove the sensitive information from the learned representations.
> > To poison this learning algorithm, we propose **F-Attack-DNN (FAD)** which is a simple generalization of F-Attack. In FAD, it also solves similar overall optimization objective as in Eq.(3) in Section 3. The only difference is: in the outer-minimization, FAD imitates the (re)-training process of fair training [8] to update the model. In fact, the only difference between FAD and F-Attack is about how the outer-minimization problem is solved, which is how the fair classification model  is updated during the (re)-training process. Plz refer Appendix A.4 for more detailed discussion about the fair method we studied and our proposed FAD Attack.
> >
> > To demonstrate the effectiveness of our proposed FAD Attack, we again focus on the Adult
> > Census Dataset and consider the fairness criteria as Equalized True Positive Rate (TPR) between genders. Under this dataset, we consider two layer Multi-Layer Perceptron (MLP) for feature extraction and classification. In our experiments, we
> > insert various fraction (from 0\% to 40\%) of poisoning samples to the training set. In the following table, we report the models’ goodness of accuracy and fairness (for 10 runs), for the models which are obtained via [8] without any defenses. From the result, we find that FAD can achieve much stronger attacking effectiveness than other baseline attacks, such as label flip (FL), attribute flip (AF) and Min-Max Attack (MM) with Z = 0 or 1. This experimental result implies the generality of F-Attack and its potential to extend to various fairness training schemes.
> >
> > |              | LF    |       | AF    |       | MM0   |       | MM1   |       | FAD0      |           | FAD1      |        |
> > |--------------|-------|-------|-------|-------|-------|-------|-------|-------|-----------|-----------|-----------|--------|
> > | Ratio        | Acc   | Fair  | Acc   | Fair  | Acc   | Fair  | Acc   | Fair  | Acc       | Fair      | Acc       | Fair   |
> > | 0%           | 0.797 | 0.960 | 0.797 | 0.960 | 0.797 | 0.960 | 0.797 | 0.960 | 0.797     | 0.960     | 0.797     | 0.960  |
> > | 10%          | 0.781 | 0.972 | 0.795 | 0.960 | 0.774 | 0.970 | 0.778 | 0.960 | **0.757** | **0.936** | 0.775     | 0.954 |
> > | 20%          | 0.783 | 0.972 | 0.798 | 0.958 | 0.776 | 0.960 | 0.740 | 0.963 | 0.731     | **0.945** | **0.713** | 0.952  |
> > | 30%          | 0.766 | 0.967 | 0.795 | 0.958 | 0.724 | 0.977 | 0.692 | 0.967 | 0.6767    | **0.947** | **0.644** | 0.965  |
> > | 40%          | 0.741 | 0.971 | 0.797 | 0.936 | 0.701 | 0.971 | 0.668 | 0.987 | 0.655     | **0.924** | **0.647** | 0.961  |
> >
> >
> >
> > **Q4. Should there be an $\leq$ instead of $=$, between Eq.(2) and Eq.(3) in our paper?**
> >
> > We apologize for the possible unclearness. In Section 2 of our paper, we mention ``we define the empirical loss function as $L(D, w)$ as the average loss value of the model'', which is the sum of training loss divided by the number of training samples. Given that the sets $D_C$ and $D_P$ have the size $n$ and $\epsilon \cdot n$, we have the following calculations (we use $l(x_i)$ to represent the model loss on a specific sample $x_i$):
> >
> > (1) $L(D_C, w) = \frac{1}{n} \sum_{i \in D_C} l(x_i)$
> >
> > (2) $\epsilon \cdot L(D_P, w) = \epsilon \cdot  \frac{1}{\epsilon \cdot n} \sum_{i \in D_P} l(x_i) = \frac{1}{n} \sum_{i \in D_P} l(x_i)$
> >
> > Combine Eq.(1) and Eq.(2)
> >
> > $L(D_C, w)+ \epsilon \cdot L(D_P, w) = \frac{1}{n} \sum_{i \in D_C \cup D_P} l(x_i) = (1+\epsilon) L(D_C \cup D_P)$
> >
> >
> > **References**
> >
> > [1] FR-Train: A Mutual Information-Based Approach to Fair and Robust Training, Roh et al, ICML 2020
> >
> > [2] Fair Classification with Group-Dependent Label Noise, Wang et al, FAccT, 2021
> >
> > [3] Robust optimization for fairness with noisy protected groups. Wang et al, NeurIPS 2020
> >
> > [4] Noise-tolerant fair classification, Wang et al, NeurIPS 2019
> >
> > [5] Fair Classification with Adversarial Perturbations, Celis et al, NeurIPS 2021
> >
> > [6] Towards Poisoning of Deep Learning Algorithms with
> > Back-gradient Optimization, Muñoz-González et al.
> >
> > [7] Certified Defenses for Data Poisoning Attacks, Steinhardt et al, NeurIPS 2017
> >
> > [8] Censoring representations with an adversary. Edwards \& Storkey et al, ICLR 2016

---

### Author Response · Authors · 2022-11-20
**General Response to all reviewers**

Thank you all reviewers for reviewing our paper. All changes in the revised version are marked in blue in the updated submission.

* In the revised version of our paper (in appendix A.4), we include one more empirical study in DNN-based fair classification model, which demonstrate that our proposed attack method and the conclusions are **general** and **representative**.

* We also add more visualizations on the poisoning scores calculated by our defense method RFC, which demonstrate the ability of RFC to find abnormal samples.

---

### Decision · Program_Chairs · 2023-01-20

**Decision:**

Reject

**Justification For Why Not Higher Score:**

Little special insight obtained by studying the intersection of poisoning and fairness.

**Justification For Why Not Lower Score:**

N/A

**Metareview: Summary, Strengths And Weaknesses:**

The authors studying poisoning attacks which are effective specifically against fair classification methods. The hope would be that there would be some special insights that arise that arise as a result of combining poisoning and fairness, but it seems relatively limited. Specifically, it seems that a straightforward modification of a prior method of Munoz-Gonzalez et al., to handle fairness constraints. Study is also restricted to relatively simple settings. On the positive side, the paper was relatively thorough. On the whole though, the paper wasn't considered strong enough to warrant acceptance.